# Targeting Gα$_{13}$-integrin interaction ameliorates systemic inflammation

Ni Cheng [1], Yaping Zhang [1], M. Keegan Delaney[1,2], Can Wang [1], Yanyan Bai [1], Randal A. Skidgel[2] & Xiaoping Du [1] ✉

Systemic inflammation as manifested in sepsis is an excessive, life-threatening inflammatory response to severe bacterial or viral infection or extensive injury. It is also a thrombo-inflammatory condition associated with vascular leakage/hemorrhage and thrombosis that is not effectively treated by current anti-inflammatory or anti-thrombotic drugs. Here, we show that MB2mP6 peptide nanoparticles, targeting the Gα$_{13}$-mediated integrin "outside-in" signaling in leukocytes and platelets, inhibited both inflammation and thrombosis without causing hemorrhage/vascular leakage. MB2mP6 improved mouse survival when infused immediately or hours after onset of severe sepsis. Furthermore, platelet Gα$_{13}$ knockout inhibited septic thrombosis whereas leukocyte Gα$_{13}$ knockout diminished septic inflammation, each moderately improving survival. Dual platelet/leukocyte Gα$_{13}$ knockout inhibited septic thrombosis and inflammation, further improving survival similar to MB2mP6. These results demonstrate that inflammation and thrombosis independently contribute to poor outcomes and exacerbate each other in systemic inflammation, and reveal a concept of dual anti-inflammatory/anti-thrombotic therapy without exacerbating vascular leakage.

[1] Department of Pharmacology, University of Illinois at Chicago College of Medicine, Chicago, IL, USA. [2] DuPage Medical Technology, Inc., Chicago, IL, USA. ✉email: xdu@uic.edu

Systemic inflammatory response syndrome can be triggered by a bacterial infection (such as sepsis)[1] or by viral infection. Under these conditions, leukocytes and macrophages release large amounts of cytokines (cytokine storm), oxidants, and other pro-inflammatory factors, which not only cause systemic inflammatory responses and vascular leakage/hemorrhage, but also thrombosis. Inflammation, vascular leakage, and thrombosis together lead to multi-organ dysfunction including acute respiratory distress syndrome (ARDS) and kidney failure, disseminated intravascular coagulation (DIC), and circulatory system collapse resulting in morbidity and mortality[1–4]. According to the Centers for Disease Control, one in three patients who die in a hospital have sepsis and at least 1.7 million adults develop sepsis annually in the U.S. which is fatal in ~270,000 cases. Similarly, viral infection can also be associated with systemic inflammation[5,6]. Thus, there are urgent needs for new drugs to treat systemic inflammation. However, there is no effective drug treatment for systemic inflammation, including that coming from severe sepsis[1–4], despite intensive research and numerous clinical trials. Most of the failed efforts in anti-sepsis drug development have concentrated on anti-inflammatory/immune regulatory therapies[7], suggesting that there are possible inflammation-independent factors contributing to poor outcomes. Interestingly, the only novel drug approved (in 2001) for the treatment of severe sepsis is the recombinant activated protein C (APC, Xigris®), which inhibits thrombin generation, thrombosis, and certain aspects of inflammation[3,8–10]. However, the clinical trials indicating improved outcomes in APC-treated septic patients also revealed a significant increase in adverse bleeding events[11]. Post-market clinical studies further revealed significant adverse effect of bleeding in both adult and pediatric patients[12], outweighing the therapeutic effects. Subsequently, a repeat clinical trial showed no adverse effect of bleeding but also no efficacy, leading to its withdrawal from the market[8,13]. These clinical trials suggest a potential benefit for antithrombotic drugs in sepsis treatment, which is consistent with the numerous data showing a close association between inflammation and thrombosis, and with the reported beneficial effects of anti-platelet drugs in treating sepsis in patients and animal models[3,8,14–19]. These studies also suggest that the effective APC treatment is associated with adverse effect of hemorrhage, which counteracts the beneficial effect of this drug. Based on these basic and clinical studies, we hypothesize that inflammation, vascular leakage/hemorrhage, and thrombosis independently contribute to poor outcomes in systemic inflammation, in addition to interdependently exacerbating each other. This hypothesis also suggests that it would be more effective to target both inflammation and thrombosis without exacerbating vascular leakage/hemorrhage in treating systemic thrombo-inflammatory conditions.

The integrin family of cell adhesion receptors plays important roles in the functions of platelets and leukocytes[20–22]. We have shown previously that Gα13 directly interacts with an ExE motif in the cytoplasmic domains of integrin β3 subunits and this binding is selectively important in stimulating integrin outside-in signaling, without affecting inside-out signaling and the ligand-binding function of integrin αIIbβ3[23,24]. Gα13-integrin interaction is important in thrombosis but apparently dispensable for hemostasis[23], and peptides derived from the β3 ExE motif potently inhibited thrombosis without affecting hemostasis in mouse models[23,25]. A form of the ExE motif is also present in leukocyte β2 integrins[23], which are critically important in inflammation[26].

In this work, we explore the effect of a peptide (MB2mP6) derived from the β2 ExE motif and the effects of leukocyte and platelet Gα13 knockout on systemic thrombo-inflammatory conditions using a mouse sepsis model induced by cecal ligation-puncture (CLP). Our data demonstrate that MB2mP6 inhibits Gα13-integrin interaction and integrin outside-in signaling of both β2 integrins in leukocytes and β3 integrins in platelets without affecting inside-out signaling and adhesion function of leukocytes and platelets. Importantly, MB2mP6 inhibits both microvascular thrombosis and inflammatory cytokine release and improves survival rates in the CLP-induced sepsis model in mice. Furthermore, platelet selective knockout of Gα13 inhibits thrombosis with only moderate effect on inflammation, whereas leukocyte-selective knockout of Gα13 inhibits inflammation with minimal effect on thrombosis. Leukocyte- or platelet- selective knockout of Gα13 only moderately increases the survival rate of septic mice. In contrast, dual leukocyte and platelet knockout of Gα13 inhibits both thrombosis and inflammation and markedly improves the survival rate of mice in CLP sepsis, similar to that found with MB2mP6. These results support the hypothesis that thrombosis and inflammation are independently and interdependently important in the outcome of the systemic thrombo-inflammatory conditions and introduce a concept of dual anti-inflammatory/anti-thrombotic therapy that does not exacerbate vascular leakage for treating systemic inflammation.

## Results

**MB2mP6 inhibits LPS-induced macrophage cytokine expression in vitro.** Integrins play critical roles in platelet and leukocyte functions. We previously discovered that ligand binding to integrin αIIbβ3 induces binding of a G-protein subunit, Gα13, to β3, transmitting outside-in signaling[20,21]. Because outside-in signaling is a post-adhesion amplification mechanism, inhibiting this process minimally affects primary integrin-mediated platelet aggregation and hemostasis[23]. Gα13 binds to a cytoplasmic ExE motif which is homologous among β2 and β3 integrins[23]. We thus designed an inhibitory peptide, MB2mP6 (Myr-FEKEKL), based on the ExE sequence of integrin β2. MB2mP6 was formulated into high-loading peptide nanoparticles (HLPN) for efficient intracellular delivery in vitro and in vivo[25]. MB2mP6 inhibited Gα13 co-immunoprecipitation with β2 integrins in LPS-stimulated macrophages differentiated from the human monocytic leukemia cell line THP-1 cells[27] (Fig. 1a). Strikingly, MB2mP6 HLPN potently inhibited LPS-induced expression of proinflammatory cytokines IL-1β, IL-6, IL-12 p40, and TNFα in THP-1-differentiated macrophages (Fig. 1b-e). MB2mP6 also inhibited LPS-stimulated cytokine IL-1β and IL-6 expression in mouse bone marrow-derived macrophages (BMDM; Supplementary Fig. 1a, b), suggesting that Gα13-integrin interaction is important in macrophage proinflammatory function. To investigate whether the inhibitory effect of MB2mP6 on cytokines is related to cell adhesion, which is known to be integrin-dependent, we compared the cytokine expression induced by LPS under conditions of adhesion and suspension. The expression of inflammatory cytokines IL-1β, IL-6, IL-12 p40, and TNFα induced by LPS was significantly lower in suspended THP-derived macrophages than in adherent cells (Fig. 1f–i), suggesting that cell adhesion greatly promotes cytokine expression. We further investigated whether MB2mP6 affects macrophage adhesion. THP-1-differentiated macrophages were pretreated with MB2mP6 HLPN or scrambled peptide control for 20 min and then loaded onto 48-well plates pre-coated with integrin ligand human (h)ICAM-1 (10 μg mL⁻¹). MB2mP6 had no effect on the adhesion of THP-1-differentiated macrophages on hICAM-1 (Fig. 1j). These data suggest that MB2MP6 inhibits post-adhesion signaling events, which is outside-in signaling, and thus macrophage cytokine expression. Leukocyte adhesion is important in phagocytosis.

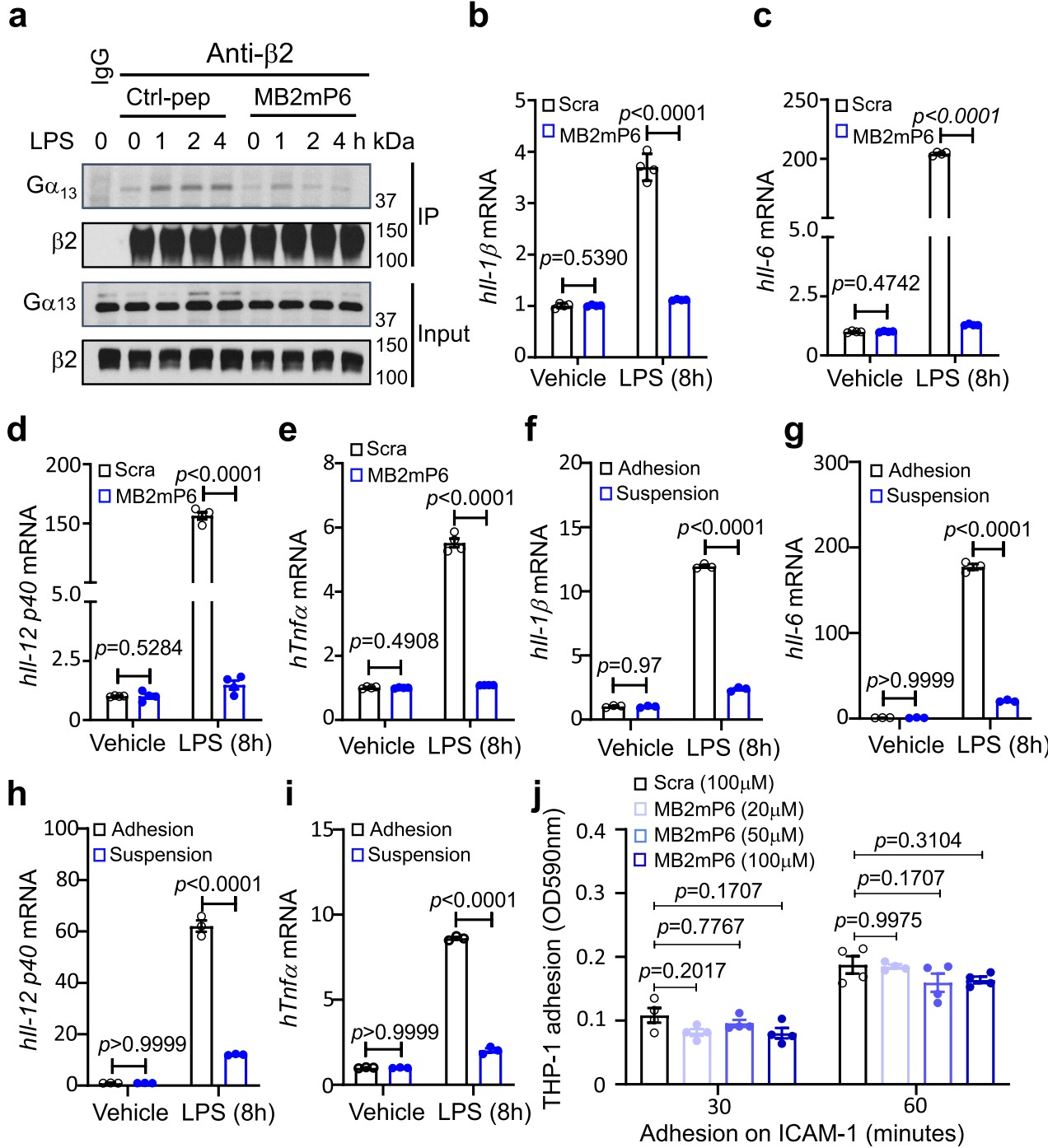

**Fig. 1 MB2mP6 blocks Gα13 interactions with β2 integrins and inhibit adhesion-dependent macrophage cytokine expression. a** Representative immunoblots (from three independent experiments) of anti-β2 antibody-co-immunoprecipitated integrin β2 subunit and Gα13 in macrophages differentiated from PMA-stimulated THP-1 cells. The macrophages were pretreated with MB2mP6 or control peptide HLPN (50 µM) for 20 min and then stimulated with LPS (100 ng mL$^{-1}$) for the indicated time prior to immunoprecipitation. Immunoprecipitates and total cell lysates were then immunoblotted with both anti-β2-integrin and anti-Gα13 antibodies. **b–e** Comparison between MB2mP6 HLPN and scrambled peptide control (Scra) in Il-1β (**b**), Il-6 (**c**), Il-12 p40 (**d**), and Tnfα (**e**) mRNA expression in THP-1-derived macrophages stimulated with 100 ng mL$^{-1}$ LPS. The mRNA expression was detected by qRT-PCR (all groups, $n = 4$, independent cultures). **f–i** Comparison between adherent and non-adherent THP-1-derived macrophages in the levels of cytokine IL-1β (**f**), IL-6 (**g**), IL-12 p40 (**h**), and TNFα (**i**) expression as detected by qRT-PCR (all groups, $n = 3$, independent cultures). **j** No effect of MB2mP6 pretreatment (20−100 µM) on adhesion of THP-1-derived macrophages onto hICAM-1 (10 µg mL$^{-1}$) precoated in 48-well plates (all groups, $n = 4$, independent cultures). All data are shown as mean ± SEM. Data in (**b–j**) were analyzed by two-way ANOVA with the post hoc Bonferroni's multiple comparisons test (**b–e**) or Sidak's multiple comparisons (**f–i**) or Tukey's multiple comparisons tests (**j**).

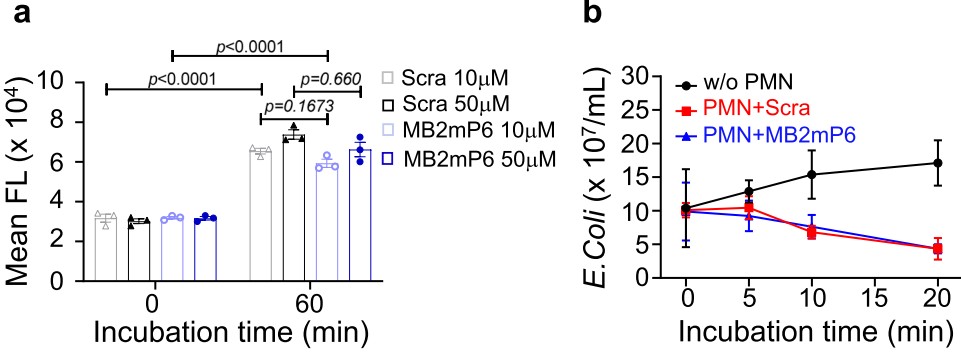

**Fig. 2 MB2mP6 does not affect phagocytosis and bacteria killing by mouse neutrophils. a** Phagocytosis of Alexa fluor 488-labeled *E. Coli* by mouse neutrophils pretreated with MB2mP6 (10 or 50 μM) as detected by flow cytometry (all groups, $n = 3$, independent isolated neutrophils). **b** In vitro *E. coli* (K-12 strain) killing by mouse neutrophils pretreated with MB2mP6 or scrambled control peptide (50 μM) was detected at various time points (all groups, $n = 4$, independent isolated neutrophils). Data are shown as mean ± SEM. Data were analyzed by two-way ANOVA with the post hoc Tukey's multiple comparisons tests.

Thus, we also tested whether MB2mP6 affects phagocytic function. To our surprise, MB2mP6 did not affect phagocytosis of Alexa fluor 488-labeled *E. Coli* (Fig. 2a) and bacteria killing (Fig. 2b) by mouse neutrophils, suggesting that the Gα13-integrin interaction may not be critical for the phagocytic function of neutrophils.

**MB2mP6 inhibits integrin outside-in signaling in platelets.** To test whether MB2mP6 also inhibits Gα13-dependent outside-in signaling of the platelet integrin αIIbβ3, Gα13 was co-immunoprecipitated with β3 integrins in thrombin-stimulated human platelets that had been preincubated with MB2mP6 or control peptide. MB2mP6 inhibited Gα13-β3 integrin interaction (Fig. 3a). MB2mP6 also dose-dependently inhibited human platelet granule secretion and secretion-dependent secondary aggregation induced by low-dose thrombin (Fig. 3b, c). However, MB2mP6 had no effect on platelet aggregation (Fig. 3d) or ATP secretion (Fig. 3e) induced by high concentrations of thrombin nor on fibrinogen binding to platelets induced by PAR4-agonist peptide (Fig. 3f, g). MB2mP6 also did not affect ADP-induced platelet aggregation (Fig. 3h), which is minimally dependent on outside-in signaling in the presence of physiological calcium concentration. These data suggest that MB2mP6 does not directly affect inside-out signaling nor the ligand binding to integrin αIIbβ3, which mediates primary aggregation, but selectively inhibits the secondary amplification of platelet aggregation, which is mediated by outside-in signaling. Interestingly, whereas MB2mP6 HLPN did not significantly affect ATP secretion induced by a high dose of thrombin (Fig. 3e), the platelet surface P-selectin expression was still inhibited by MB2mP6 at both low and high dose thrombin stimulation (Fig. 3i).

**MB2mP6 inhibits thrombosis but does not cause bleeding in vivo.** We used the ferric chloride-induced mouse carotid artery thrombosis model to determine the in vivo effects of MB2mP6 in inhibiting thrombosis. MB2mP6 effectively inhibited FeCl3-induced mouse carotid artery thrombosis in vivo, although the anti-thrombotic effect appeared to be less potent than the β3-derived M3mP6 peptide (Fig. 4a). Importantly, MB2mP6 neither affected injury-induced tail-bleeding time (Fig. 4b), nor inflammation-induced hemorrhage in the reverse passive Arthus (rpA) reaction in mice (Fig. 4c). These data demonstrate that MB2mP6 is anti-inflammatory and anti-thrombotic but does not cause or exacerbate hemorrhage.

**MB2mP6 enhances survival in a CLP severe sepsis model.** We used the mouse cecal ligation and puncture (CLP) model of severe polymicrobial sepsis to test the possible therapeutic effects of MB2mP6 on systemic inflammation. MB2mP6 or a scrambled control peptide HLPN were infused i.v. immediately after CLP at a rate of $1.25 \, \mu mol \, kg^{-1} \, h^{-1}$ through a pre-placed jugular vein cannula (Fig. 5a). To mimic clinical treatment, antibiotic therapy (Claforan solution) was also started at the same time with both groups. The MB2mP6 group showed a significant improvement in survival probability (survival rate 71% in the MB2mP6 group vs 26% in the control peptide group at 192 h (8 days); $p = 0.002$) (Fig. 5a). The control peptide group was similar to the saline treatment group (Fig. 5a). To more closely mimic the clinical conditions in sepsis treatment, we started MB2mP6 infusion 6 or 18 h after CLP onset. When started at 6 h after CLP, MB2mP6 also significantly improved the 8-day survival probability (survival rate 42% in the MB2mP6 group vs 6.25% in the control group, $p = 0.003$, Fig. 5b). Even when started at a very late 18 h after CLP onset, MB2mP6 treatment still resulted in significant improvement in survival probability compared to a scrambled peptide control group (survival rate 6.7% in the MB2mP6 group vs 0% in the control group, $p = 0.011$, Fig. 5c), although the percentages of survival further decreased in both MB2mP6 and control groups. The decrease in survival rate at 18 h was likely caused by the delayed start of antibiotic treatment, which was in the treatment regimen together with MB2mP6 or control peptide HLPN. In testing the inflammation and coagulation states of the septic mice at the time of treatment after starting CLP, we found that the proinflammatory cytokines, IL-6, and even the late phase cytokine TNFα, significantly increased at 6 h (Supplementary Fig. 2a, b). At 18 h, IL-6 had already decreased to close to base line level, whereas TNFα further increased. The "anti-inflammatory" cytokine IL-10 was near the baseline at 6 h but significantly increased at 18 h suggesting entry into the severe "suppression" phase at this late time point (Supplementary Fig. 2c). Fibrinogen and thrombin-anti-thrombin complex (TAT) had already increased at 6 h and further increased at 18 h, suggesting "hypercoagulant" state (Supplementary Fig. 2d, e). The fibrinolytic product D-dimer only became elevated at 18 h (Supplementary Fig. 2f), consistent with its association with a more grave stage of sepsis. These results suggest that systemic inflammation had fully developed at 6 h after sepsis onset, and reached the late severe stage associated with DIC at 18 h. Even at these late phases, MB2mP6 treatment still had a beneficial effect. Thus, MB2mP6 effectively treats and improves the survival probability of severely septic mice.

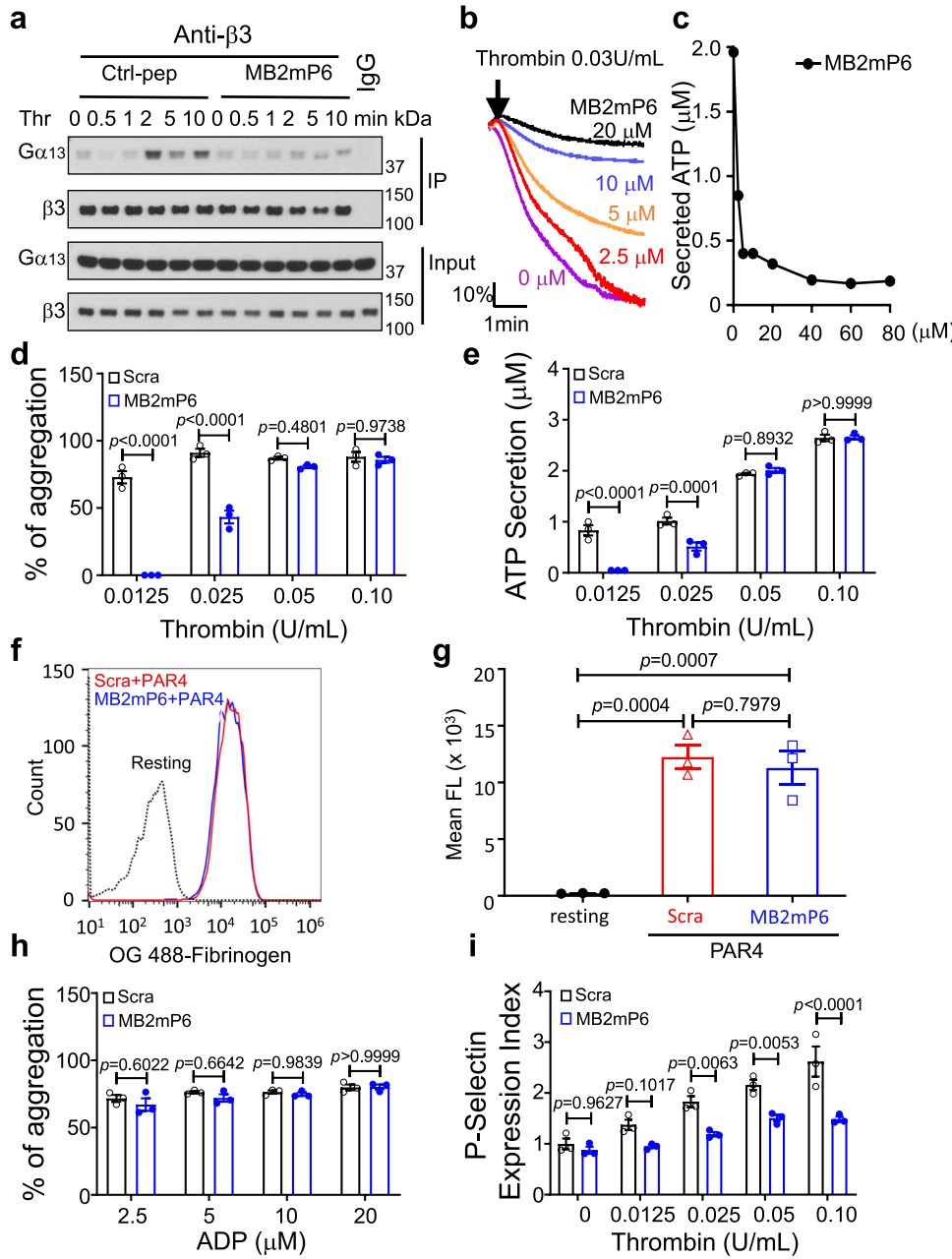

**Fig. 3 The inhibitory effects of MB2mP6 on G$\alpha_{13}$ interactions with $\beta_3$ integrins and on platelet function. a** Representative immunoblots (from three independent experiments) of co-immunoprecipitation of $\beta_3$ integrin and G$\alpha_{13}$ with anti-$\beta_3$ antibody in human platelets pretreated with MB2mP6 or control peptide HLPN (50 µM) for 5 min and then stimulated with thrombin (0.025 U mL$^{-1}$) for various lengths of time. The immunoprecipitates and total cell lysates were immunoblotted with both anti-$\beta_3$-integrin and anti-G$\alpha_{13}$ antibodies respectively. **b** Representative thrombin-induced human platelet aggregation tracing showing a dose-dependent inhibition of aggregation by MB2mP6 HLPN. **c** Representative study showing 0.03 U mL$^{-1}$ thrombin-induced ATP secretion in washed human platelets is dose-dependently inhibited by MB2mP6 HLPN. **d** Effects of MB2mP6 HLPN on thrombin-induced mouse platelet aggregation induced by increasing doses of thrombin (all groups, $n = 3$, independent isolated platelets). **e** Effects of MB2mP6 HLPN on ATP secretion from mouse platelets stimulated with increasing doses of thrombin (all groups, $n = 3$, independent isolated platelets). **f** Effects of MB2mP6 HLPN on 500 µM PAR4AP–induced binding of Oregon Green-labeled fibrinogen to mouse platelets. **g** Quantification of the binding of Oregon Green-labeled fibrinogen to mouse platelets induced by PAR4AP (500 µM) pretreated with MB2mP6 or scrambled peptide HLPN (all groups, $n = 3$, independent isolated platelets). **h** Lack of effect of MB2mP6 HLPN on ADP-induced mouse platelet aggregation in presence of 100 µg mL$^{-1}$ fibrinogen (all groups, $n = 3$, independent isolated platelets). **i** Effects of MB2mP6 HLPN on P-selectin expression in mouse platelets induced by various dose of thrombin (all groups, $n = 3$, independent isolated platelets). All data are shown as mean ± SEM. Data in (**d**), (**e**), (**h**), and (**i**) were analyzed by two-way ANOVA with the post hoc Sidak's multiple comparisons test; Data in (**g**) was analyzed with one-way ANOVA with Tukey's multiple comparisons test.

**MB2mP6 inhibits inflammation and thrombosis during sepsis.**
To assess the effect of MB2mP6 on inflammation during sepsis in vivo, we tested pro-inflammatory cytokine levels in mouse serum collected 24 h after CLP. Control mice showed significantly increased secretion of IL-6 and TNFα, which were significantly reduced by MB2mP6 treatment (Fig. 5d, e). Similarly, expression of IL-6 and TNFα transcripts in septic mouse lung at 24 h were also induced by CLP and significantly suppressed by MB2mP6

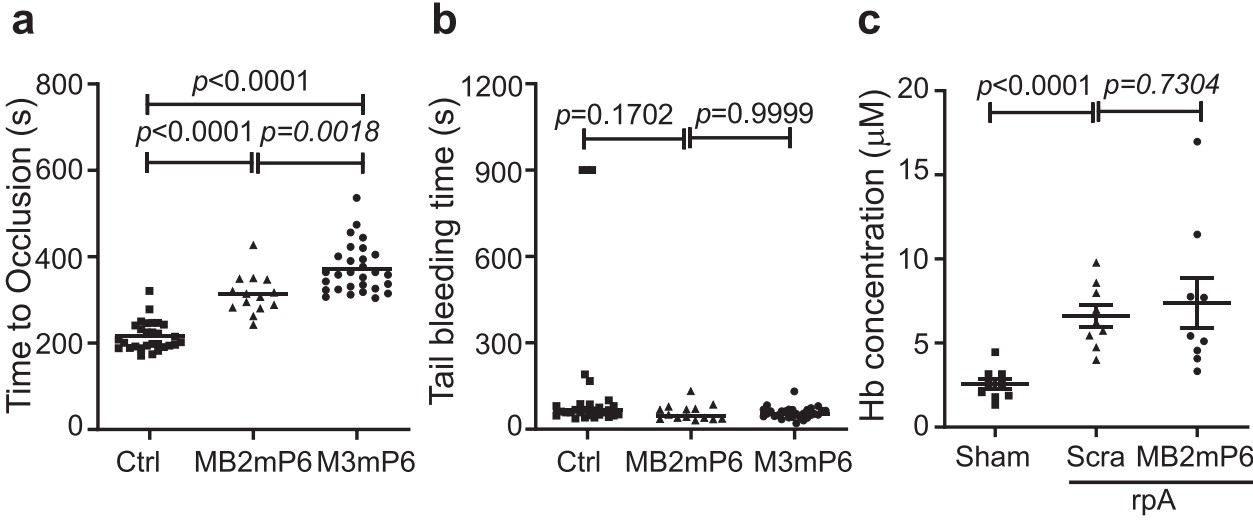

**Fig. 4 The effects of MB2mP6 on FeCl₃-induced carotid artery thrombosis and on hemorrhage in vivo. a**, **b** Comparison of the effects of MB2mP6 and M3mP6 (Myr-FEEERL) on FeCl₃-induced carotid artery thrombosis (**a**) and tail-bleeding time (**b**) (control, $n = 29$; MB2mP6, $n = 14$; M3mP6, $n = 29$, independent animals). **c** Comparison between MB2mP6 and scrambled peptide on inflammation-induced hemorrhage in the rpA assay ($n = 9$ for all groups, independent animals). All data are shown as mean ± SEM. Data in (**a**), (**b**) and (**c**) were analyzed by unpaired Mann–Whitney test (two-tailed).

infusion (Supplementary Fig. 3a, b). These data demonstrate that MB2mP6 has potent anti-inflammatory effects in vivo during severe sepsis.

Severe sepsis causes microvascular thrombosis in kidney glomeruli, damaging kidney function[28,29]. Indeed, in CLP septic mice, we observed microvascular thrombosis in kidney glomeruli, as indicated by Mallory's phosphotungstic acid hematoxylin (PTAH) staining (Fig. 5f), anti-fibrin staining of fibrin deposition, anti-$\alpha_{IIb}$ staining of platelets, and anti-VWF staining (Supplementary Fig. 4a–c). CLP sepsis also impaired renal function as indicated by elevation of BUN (Fig. 5g), creatinine, and cystatin C (Supplementary Fig. 4d, e) in the control group 24 h after CLP. MB2mP6 treatment significantly inhibited glomerular thrombosis and reduced BUN, creatinine, and cystatin C (Fig. 5f, g and Supplementary Fig. 4a–e). MB2mP6 also decreased the elevated TAT levels without affecting fibrinogen concentrations in the whole blood of septic mice 24 h after treatment (Supplementary Fig. 5a, b). Thus, MB2mP6 inhibits both inflammation and thrombosis and improves renal function in septic mice.

**MB2mP6 reduces septic lung vascular leak and does not cause bleeding.** An important cause of mortality in sepsis and other systemic and pulmonary inflammatory conditions is the ARDS, in which inflammation-induced vascular leakage (hyperpermeability) and hemorrhage acutely cause severe lung edema, impairing respiratory function. To test whether MB2mP6 protected from lung vascular leakage in septic mice, we examined the lung permeability 24 h after CLP. MB2mP6 treatment significantly reduced Evan's blue infiltration into the lung as compared with control peptide when treatment started immediately after CLP (Fig. 6a) and when treatment started 6 h after CLP (Fig. 6b). Thus, MB2mP6 alleviates the vascular leakage in septic lungs. To investigate the effects of MB2mP6 on bleeding during CLP-induced sepsis, we measured the hemoglobin (Hb) contents in the stools collected from septic mice 24 after CLP onset. Indeed, hemoglobin levels in the stools of septic mice were elevated 24 h after CLP surgery compared to before surgery, and the Hb level was not affected by MB2mP6 treatment as compared with scrambled peptide control (Supplementary Fig. 6). Taken together, these data suggest that MB2mP6 attenuates vascular

hyperpermeability/vascular leakage during severe sepsis with no adverse effect on bleeding. These results indicate that MB2mP6 may be potentially effective in preventing ARDS in systemic inflammation.

**Effects of selective Gα₁₃ knockout in leukocytes or platelets on sepsis.** The beneficial effects of MB2mP6 in treating sepsis suggest that Gα₁₃-integrin interaction in either leukocytes or platelets or both are important in the development of systemic inflammation. To test the importance of leukocyte Gα₁₃, we generated leukocyte-specific Gα₁₃ knockout mice by mating Gα₁₃^{fl/fl} mice generously provided by Dr. Stefan Offermanns, Max Planck Institute for Heart and Lung Research, Germany[30], with LysM-Cre mice. The western blotting analysis confirmed the absence of Gα₁₃ protein in both macrophages and neutrophils but not platelets of Gα₁₃^{fl/fl-LysMCre} mice (Supplementary Fig. 7a, b). Leukocyte-specific Gα₁₃ knockout (Gα₁₃^{fl/fl-LysMCre}) mice had a moderately but significantly improved 8-day survival rate in CLP-induced sepsis as compared with control groups (44% in Gα₁₃^{fl/fl-LysMCre} group vs 20% in the LysM-Cre only group vs 19% in the Gα₁₃^{fl/fl} control group, $p = 0.044$ between Gα₁₃^{fl/fl-LysMCre} and LysM-Cre) (Fig. 7a). However, the survival rate in leukocyte-specific Gα₁₃-knockout mice was significantly lower than that in the MB2mP6-treated mice (Fig. 7a vs Fig. 5a and summarized in Supplementary Fig. 8, $p < 0.05$) Importantly, the elevation of serum cytokines IL-6 and TNFα levels 24 h after CLP were significantly inhibited in Gα₁₃^{fl/fl-LysMCre} mice (Fig. 7b, c), suggesting that leukocyte Gα₁₃ plays a major role in the inflammatory state induced by sepsis. In contrast, glomerular microvascular thrombosis as indicated by fibrin deposition was slightly but not significantly decreased in Gα₁₃^{fl/fl-LysMCre} mice compared with control Gα₁₃^{fl/fl} mice (Fig. 7d) and the elevation of the kidney damage marker BUN was also not reduced in the blood of Gα₁₃^{fl/fl-LysM-Cre} septic mice (Fig. 7e). These data suggest that leukocyte Gα₁₃ is not a critical factor in kidney microvascular thrombosis and damage and that the protective effect of leukocyte-specific Gα₁₃ knockout on septic systemic inflammation was mainly due to inhibition of leukocyte-mediated cytokine secretion and inflammation.

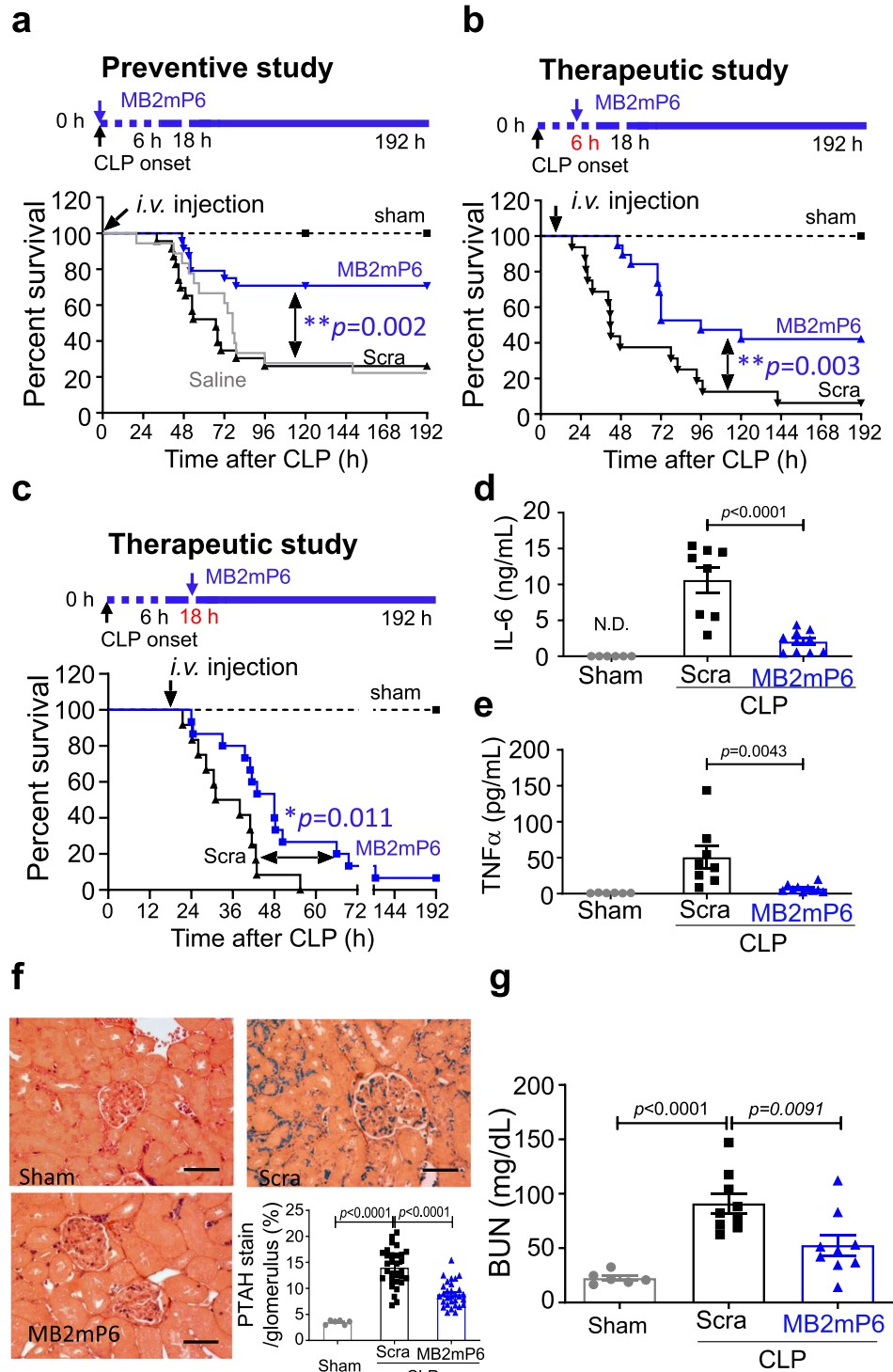

To assess the contribution of platelet Gα13 to the therapeutic effect of MB2mP6, and its role in systemic inflammation, platelet-specific knockout mice were generated by mating Gα13^{fl/fl} mice with PF4-Cre mice (Supplementary Fig. 7a, b). The 8-day (192 h) survival rate in the platelet-specific Gα13 knockout (Gα13^{fl/fl-PF4Cre}) mice after CLP sepsis was moderately but significantly improved compared with control PF4-Cre or Gα13^{fl/fl} mice (39.2% vs 7.6% in the PF4-Cre control group; $p = 0.001$) (Fig. 7f), but significantly lower than MB2mP6-treated mice (Fig. 5a vs Fig. 7f and summarized in Supplementary Fig. 8, $p < 0.05$). In contrast to leukocyte-specific Gα13 knockout, the CLP-induced

glomerular microvascular thrombosis and kidney function impairment (indicated by blood BUN levels 24 h after CLP) were both markedly reduced in Gα13^{fl/fl-PF4Cre} mice compared with Gα13^{fl/fl} mice (Fig. 7g, h). Thus platelet Gα13 plays an important role in glomerular microvascular thrombosis and kidney injury during sepsis, and is likely responsible for the therapeutic effect of MB2mP6 on glomerular thrombosis and kidney injury. However, the serum level of the inflammatory cytokine TNFα was not significantly reduced in Gα13^{fl/fl-PF4Cre} mice (Fig. 7i), although cytokine IL-6, levels in mouse serum were partially (and significantly) reduced (Fig. 7j). Thus, it appears that platelet-

**Fig. 5 MB2mP6 improves survival in a CLP sepsis model and reduces inflammatory cytokines, thrombosis and prevents sepsis-induced organ injury.**
**a** Effect of MB2mP6 and scrambled peptide (Scra) treatment immediately after CLP surgery on survival of C57BL/6 septic mice (Sham, $n = 6$; Scra, $n = 23$; MB2mP6, $n = 24$, independent animals). Log-rank (Mantel–Cox) test using GraphPad Prism software (two-tailed). **b** Effect of MB2mP6 and scrambled peptide (Scra) treatment 6 h after CLP onset on survival of C57BL/6 septic mice (sham, $n = 6$; Scra, $n = 16$; MB2mP6, $n = 19$, independent animals). Log-rank (Mantel-Cox) test using GraphPad Prism software (two-tailed). **c** Effect of MB2mP6 and scrambled peptide (Scra) treatment 18 h after CLP onset on survival of C57BL/6 septic mice (Sham, $n = 6$; Scra, $n = 12$; MB2mP6, $n = 15$, independent animals). Log-rank (Mantel-Cox) test using GraphPad Prism software (two-tailed). In (**a**), (**b**), and (**c**), "Sham" indicates mice receiving sham surgery (without CLP). **d, e** MB2mP6 inhibits protein levels of cytokines IL-6 and TNFα in mouse serum measured by ELISA 24 h after CLP (sham, $n = 6$; Scra, $n = 8$; MB2mP6, $n = 10$, independent animals). **f** Representative images of fibrin deposition detected by PTAH staining in mouse kidney glomeruli 24 h after CLP. Bars indicate the percentage of PTAH positive stained area per glomerulus (30 random glomeruli from 6 mice/group) (sham, $n = 6$; Scra, $n = 30$; MB2mP6, $n = 30$, independent glomeruli). Scale bar = 100 µm. **g** Plasma level of BUN in septic mice 24 h after CLP: comparison between MB2mP6-treated mice and scrambled peptide-treated mice (sham, $n = 6$; Scra, $n = 9$; MB2mP6, $n = 9$, independent animals). All data are shown as mean ± SEM. Data in (**d**) were analyzed by student *t*-test (two-tailed); data in (**e**) and (**g**) were analyzed by one-way ANOVA with post hoc Tukey's multiple comparisons test; data in (**f**) were analyzed by a two-tailed Mann–Whitney test.

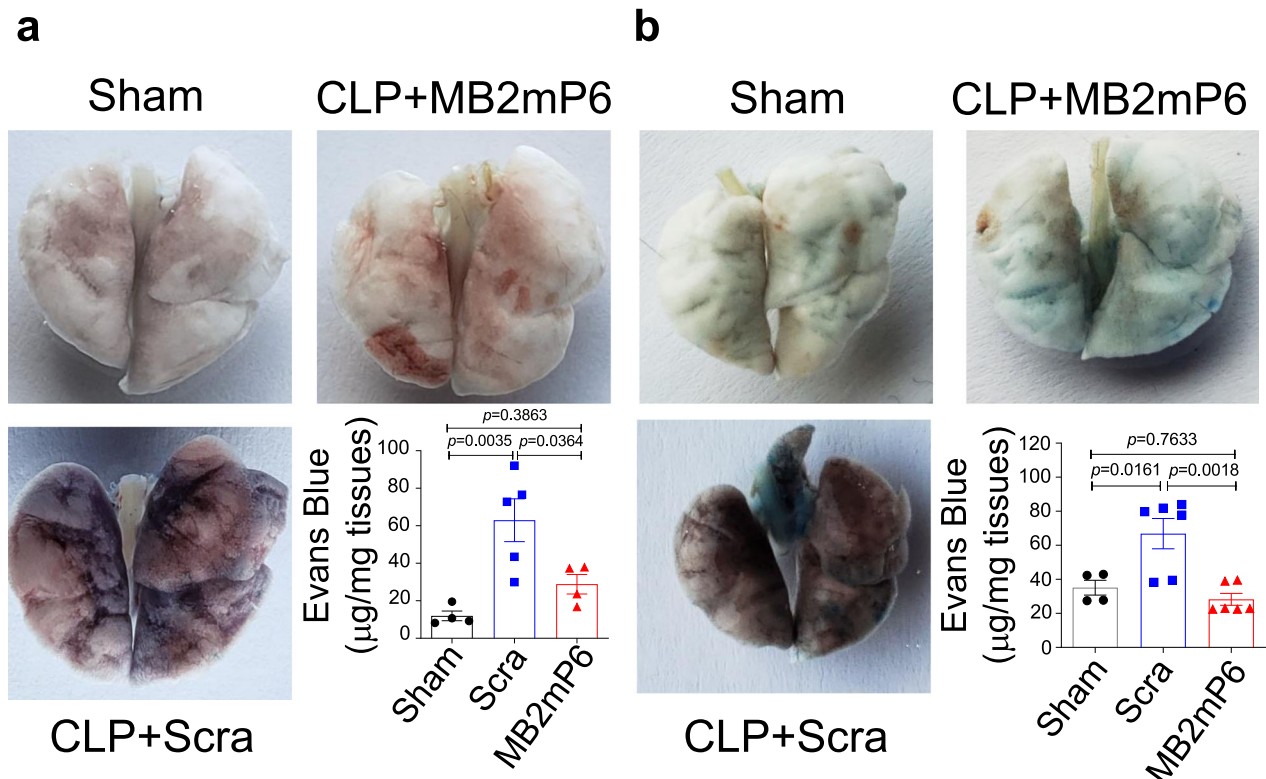

**Fig. 6 MB2mP6 inhibits CLP-induced lung vascular leakage induced by CLP sepsis. a** Representative images of septic mouse lungs 24 h after CLP. The mice were treated with scrambled peptide (Scra) or MB2mP6 immediately after CLP onset. At 23.5 h after CLP, the mice were i.v. injected with Evans blue albumin (EBA; 1%, 25 mg kg$^{-1}$ body weight). After 30 min, the mouse lungs were harvested, and the Evans blue was extracted by incubating with formamide at 65 °C for 18 h. The EBA contents in the lung tissues were calculated using a standard EBA curve detected at 620 nm absorbance with 740 nm reference (sham, $n = 4$; control, $n = 5$; MB2mP6, $n = 4$, independent animals). **b,** Representative images of septic mouse lungs 24 h after CLP treated with Scrambled peptide or MB2mP6 6 h after CLP onset. The Evans blue contents in septic mouse lung 24 h after CLP were analyzed as in (**a**) (sham, $n = 4$; control, $n = 6$; MB2mP6, $n = 6$, independent animals). All data are shown as mean ± SEM. Data were analyzed by one-way ANOVA with Tukey's multiple comparisons test.

specific Gα$_{13}$ knockout moderately reduced mortality in septic mice mainly by inhibiting microvascular thrombosis, although platelet Gα$_{13}$ may also contribute to the exacerbation of the inflammatory state.

**The effect of dual leukocyte and platelet Gα$_{13}$ knockout on sepsis.** To further determine whether Gα$_{13}$ in platelets and in leukocytes has independent roles in sepsis-induced mortality and organ damage, we generated platelet/leukocyte dual Gα$_{13}$ knockout mice (Gα$_{13}$$^{fl/fl-LysM/PF4}$ double Cre mice, Supplementary Fig. 7a, b). The 8-day survival rate after CLP in the dual Gα$_{13}$ knockout mice was 66% (Fig. 8a, $p = 0.0072$ vs 29.4%

for LysM + PF4 double Cre mice; $p = 0.0001$ vs 5.9% for Gα$_{13}$$^{fl/fl}$ mice), similar to that of septic wild type mice treated with MB2mP6 (71%; Fig. 5a). This was also significantly higher than not only the control mice but also platelet-specific Gα$_{13}$ knockout (39.2%) (Fig. 7f vs Fig. 8a and summarized in Supplementary Fig. 8, $p < 0.05$) and leukocyte-specific Gα$_{13}$ knockout mice (44%) (Fig. 7a vs Fig. 8a and summarized in Supplementary Fig. 8, $p < 0.05$). Consistent with these results, the elevation of kidney injury marker BUN (Fig. 8b) and cytokines IL-6 (Fig. 8c) or TNFα (Fig. 8d) in the blood of control mice 24 h after CLP were markedly inhibited in the dual Gα$_{13}$ knockout mice. The CLP-induced glomerular

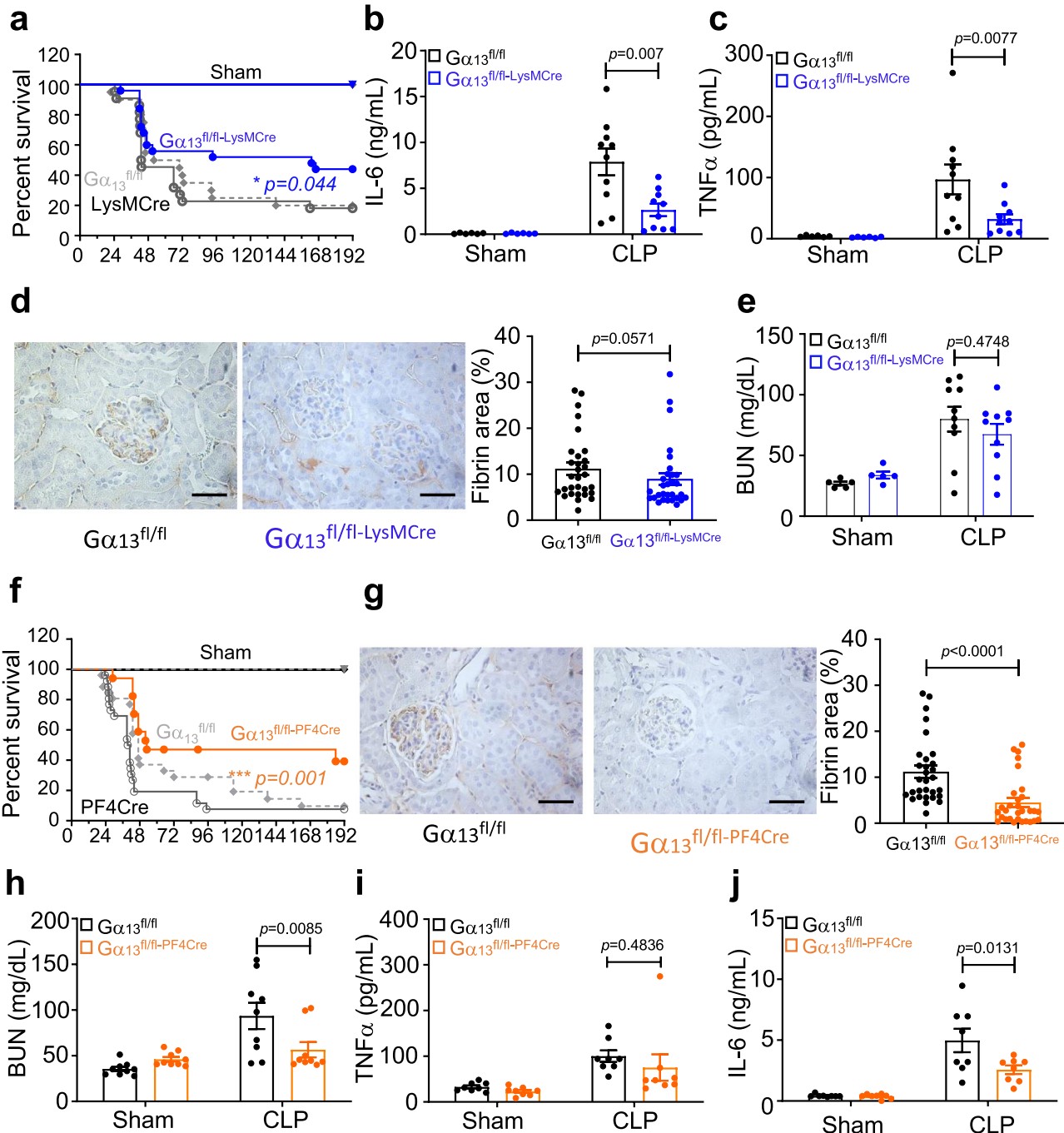

**Fig. 7 Leukocyte- or platelet-specific Gα13 knockout reduces mortality in CLP-induced septic mice. a** The survival rate of mice after CLP in Gα13fl/fl-LysMCre (leukocyte-specific Gα13 knockout) mice compared to LysM-Cre or Gα13fl/fl control mice. LysM-Cre sham, $n = 6$; LysM-Cre CLP, $n = 22$; Gα13fl/fl CLP, $n = 20$; Gα13fl/fl-LysMCre sham, $n = 6$; Gα13fl/fl-LysMCre CLP, $n = 25$. Log-rank (Mantel–Cox) test (two-tailed). **b, c** Serum levels of IL-6 and TNFα in septic mice 24 h after CLP in Gα13fl/fl-LysMCre and Gα13fl/fl control mice. All sham groups, $n = 6$; all CLP groups, $n = 10$ (independent animals). **d** Images and quantification of immunohistochemical staining of fibrin deposition in mouse kidney 24 h after CLP. Bar graph shows the percentage of fibrin-stained area per glomerulus (30 random glomeruli from 6 mice/group, all groups, $n = 30$, independent glomeruli). Scale bar = 100 μm. **e** Plasma level of BUN in septic mice 24 h after CLP in Gα13fl/fl-LysMCre mice and Gα13fl/fl mice (all sham groups, $n = 5$; all CLP groups, $n = 10$, independent animals). **f** Survival rate in mice after CLP-induced sepsis in Gα13fl/fl-PF4Cre (platelet-specific knockout) mice compared to PF4-Cre or Gα13fl/fl control mice. PF4-Cre sham, $n = 6$; PF4-Cre CLP, $n = 25$; Gα13fl/fl CLP, $n = 26$; Gα13fl/fl-PF4Cre sham, $n = 7$; Gα13fl/fl-PF4Cre CLP, $n = 17$. Log-rank (Mantel–Cox) test (two-tailed). **g** Images and quantification of immunohistochemical staining of fibrin deposition in mouse kidney 24 h after CLP. Bar graph shows the percentage of fibrin-positive stained area per glomerulus (30 random glomeruli from 6 mice/group, all groups, $n = 30$, independent glomeruli). Scale bar = 100 μm. **h,** Plasma levels of BUN in septic mice 24 h after CLP in Gα13fl/fl-PF4Cre mice compared to Gα13fl/fl mice (all groups, $n = 9$, independent animals). **i, j** Serum levels of TNFα and IL-6 in septic mice 24 h after CLP in Gα13fl/fl-PF4Cre and Gα13fl/fl control mice (all groups, $n = 8$, independent animals). All data are shown as mean ± SEM. Data in (**b**), (**c**), (**h–j**) were analyzed by two-way ANOVA with Tukey's multiple comparisons test; data in (**d**) and (**g**) were analyzed by a two-tailed Mann–Whitney test.

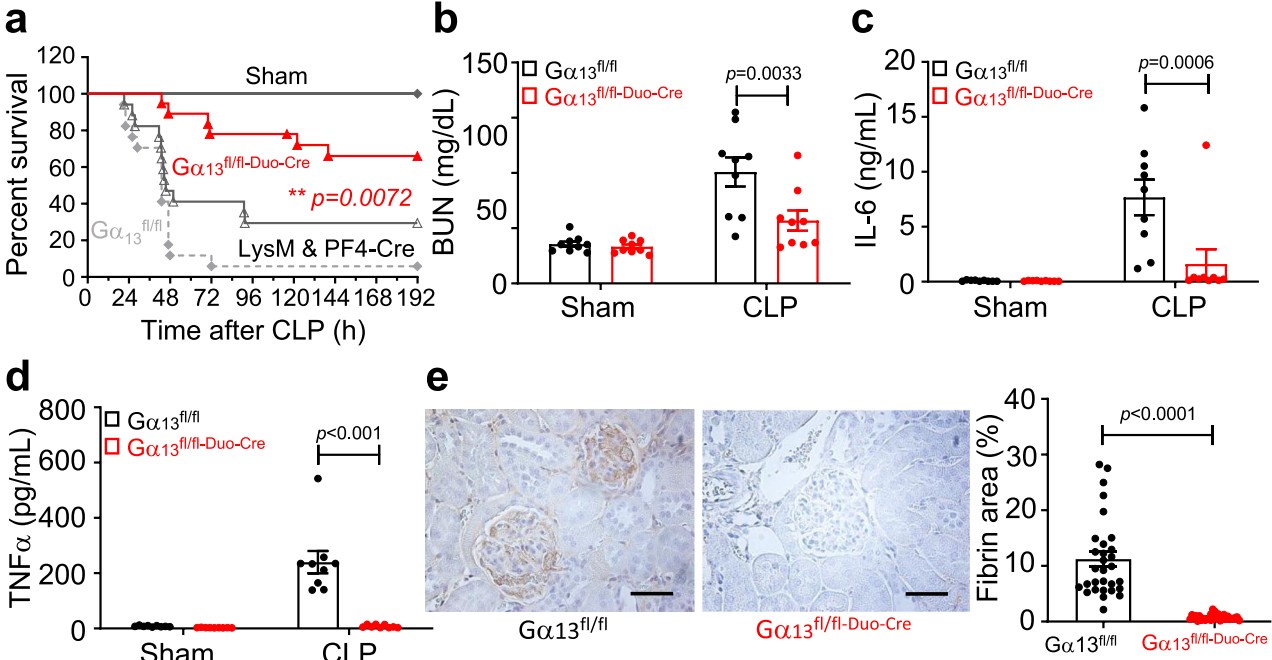

**Fig. 8 Platelet and Leukocyte dual-specific Gα₁₃ knockout reduces inflammatory cytokines, thrombosis, and renal injury and improves survival in a mouse CLP sepsis model. a** Survival rate in mice after CLP-induced sepsis in Gα₁₃$^{fl/fl-PF4/LysMCre}$ mice compared to LysM & PF4-double Cre or Gα₁₃$^{fl/fl}$ control miceDouble Cre sham, $n = 6$; double Cre CLP, $n = 16$; Gα₁₃$^{fl/fl}$ CLP, $n = 17$; Gα₁₃$^{fl/fl-PF4/LysMCre}$ sham, $n = 7$; Gα₁₃$^{fl/fl-LysMCre}$ CLP, $n = 19$ (all groups, independent animals). Log-rank (Mantel–Cox) test (two-tailed). **b** Plasma levels of BUN in septic mice 24 h after CLP in Gα₁₃$^{fl/fl-PF4/LysMCre}$ mice compared to Gα₁₃$^{fl/fl}$ mice. All groups, $n = 9$. **c, d** Serum levels of IL-6 and TNFα in septic mice 24 h after CLP in Gα₁₃$^{fl/fl-PF4/LysMCre}$ mice compared with Gα₁₃$^{fl/fl}$ control mice. All groups, $n = 9$. **e** Images and quantification of immunohistochemical staining of fibrin deposition in mouse kidney 24 h after CLP. Bar graph shows the percentage of fibrin-positive stained area per glomerulus (30 random glomeruli from 6 mice/group, all groups, $n = 30$, independent glomeruli). Scale bar = 100 µm. All data are shown as mean ± SEM. Data in (**b–d**) were analyzed by two-way ANOVA with Tukey's multiple comparisons test; data in (**e**) were analyzed by a two-tailed Mann–Whitney test.

microvascular thrombosis was also significantly reduced in the dual Gα₁₃ knockout mice as indicated by greatly reduced fibrin deposition (Fig. 8e). These data not only further support the distinct roles of Gα₁₃ in platelets and leukocytes in septic systemic inflammation, but also demonstrate that their roles in sepsis are additive. Furthermore, these data demonstrate that the effectiveness of MB2mP6 in treating septic systemic inflammation in the CLP model is likely due to its ability to inhibit Gα₁₃-integrin interaction and outside-in signaling in both platelets and leukocytes.

## Discussion

In this study, we demonstrate that Gα₁₃ and its interaction with leukocyte and platelet integrins play important roles in the thrombotic and inflammatory functions of platelets and leukocytes respectively during systemic inflammation. The leukocyte-mediated inflammatory response and platelet-dependent thrombosis both independently contribute to the poor outcome in systemic inflammation in addition to their interdependent roles in exacerbating each other as shown using the CLP model of sepsis in mice. We further show that a β₂ integrin-derived peptide, MB2mP6, by targeting and inhibiting the Gα₁₃ interaction with β₂ and β₃ integrins in leukocytes and platelets, effectively inhibits both thrombosis and inflammation to increase the survival probability of severely septic mice. The protective effect of MB2mP6 was shown even when the treatment was started in a late phase of sepsis. Importantly, MB2mP6 does not adversely affect hemostasis and even reduces pulmonary vascular leakage, a major mechanism leading to

ARDS. To our surprise, MB2mP6, while inhibiting integrin-dependent cytokine secretion, did not significantly affect the phagocytosis/killing of bacteria by leukocytes, which likely contributes to its beneficial effect on sepsis. Thus our study provides strong support for a concept of simultaneously inhibiting inflammation and thrombosis while reducing vascular leakage and maintaining hemostatic function as an effective strategy for treating systemic inflammation.

Classically, Gα₁₃ is coupled to heptahelical G protein-coupled receptors to mediate activation of the RhoA signaling pathway and contraction[31–33]. More recently, we discovered that Gα₁₃ also directly interacts with homologous ExE motifs in the cytoplasmic domains of several integrin β subunits including platelet α$_{IIb}$β₃ and β₂ in leukocytes[23,24]. Gα₁₃ binding to β₃ in platelets does not affect inside-out signaling which is important for the activation of ligand binding to integrin α$_{IIb}$β₃, nor the ligand-binding function of integrin α$_{IIb}$β₃, but selectively inhibits the ligand binding-induced outside-in signaling[23]. Mutations in the ExE motif abolishing the Gα₁₃-β₃ interaction selectively inhibited secondary amplification of platelet aggregation and cell spreading without affecting primary platelet adhesion and aggregation, which are required for primary hemostasis[23]. Thus, the β₃-derived ExE motif peptides potently inhibited thrombosis without affecting normal hemostasis, unlike all current anti-platelet drugs[23,25]. Here we further show that the β₂-derived ExE motif peptide MB2mP6 inhibits Gα₁₃ binding to β₂ integrins and potently inhibits macrophage cytokine synthesis. We also show that that cytokine expression in macrophage-like cell lines is adhesion-dependent, but MB2mP6 had no effect on adhesion per se, suggesting its effect only on outside-in signaling. Both MB2mP6 and

Gα$_{13}$ knockout in leukocytes greatly reduced cytokine levels in CLP-induced septic mice. Thus, Gα$_{13}$ is important in stimulating the cytokine storm during systemic inflammation. Importantly, MB2mP6 inhibited Gα$_{13}$-integrin interaction both in platelets and leukocytes, and inhibited both inflammation and thrombosis without affecting hemostasis. Thus, MB2mP6 has the potential to be developed into a dual anti-thrombosis/anti-inflammatory drug that does not cause vascular leakage/bleeding.

It has been recognized that severe infection or extensive injury may induce excessive inflammatory responses intertwined with vascular leakage, hemorrhage, and thrombosis (vasculopathy), which stimulate and exacerbate each other in a vicious cycle developing into a systemic thrombo-inflammatory condition with high mortality. Once this systemic response has developed, however, anti-inflammatory drugs have been proven ineffective in improving the final outcome. Our results showing independent and additive effects of leukocyte-mediated inflammation and platelet-dependent thrombosis in severe sepsis provide an explanation for past failures of anti-inflammatory drugs in treating sepsis, and importantly provide strong support for an approach of simultaneous anti-thrombotic and anti-inflammatory therapy. However, an important feature of systemic inflammation is vascular leakage and hemorrhage, which are greatly exacerbated by all current anti-thrombotic drugs and have been shown to outweigh the beneficial effect of Xigris in the previous studies[3,8]. In contrast, we demonstrate that MB2mP6, targeting the outside-in signaling of integrins both in leukocytes and platelets, does not exacerbate hemorrhage, but effectively reduces the vascular leakage in the lung. Importantly, we demonstrate that MB2mP6 is effective in treating septic systemic inflammation in the CLP mouse model, reducing both inflammation and thrombosis to enhance survival. These conceptual and translational advances thus set the stage for further studies to determine whether this new drug is effective in treating systemic inflammation in humans.

It is important to note that MB2mP6's effectiveness is not limited to preventing sepsis immediately after CLP onset, but also has significant therapeutic effects when administered 6 and even 18 h after CLP in mice. In mice, sepsis-induced increases in most cytokines as well as lung neutrophil recruitment, alveolar leak, endothelial damage, liver neutrophil and platelet recruitment with impaired sinusoidal perfusion, and acute kidney injury are all well established by 4–6 h post-CLP[34–38]. Our data showing elevation of pro-inflammatory cytokines, IL-6 and TNFα as well as the hypercoagulant state at 6 h and elevation of late anti-inflammatory cytokine IL-10 and D-dimer at 18 h, are consistent with these reports. Thus, MB2mP6 is not purely preventative, but rather disrupts the progression of systemic inflammation. However, anti-inflammatory/anti-thrombotic therapy will likely be successful if used before the grave "suppression" phase of the disease. If the patients/experimental animals have succumbed to the consequences of inflammation and thrombosis (e.g., multiple organ failure), they are unlikely to be revived with these (or likely any other) drugs. Indeed, the start of the treatment with MB2mP6/antibiotics immediately after sepsis onset resulted in better survival compared to infusion 6 or 18 h after CLP. Thus, the use of this type of drug early is likely to be more effective in disrupting the progression of systemic inflammation into the irreversible phase.

Overall, our data reveal the independent and interdependent roles of inflammation, vascular leakage/hemorrhage, and thrombosis in systemic inflammation and provides a concept and experimental drug for simultaneous anti-inflammatory and anti-thrombotic therapy without causing vascular leakage and hemorrhage. This therapeutic approach should help our defense against systemic inflammation, which is increasingly a challenge to human health.

## Methods

**Animals**. Mice used in this study were 8- to 16-weeks-old with an equal sex ratio. Gα$_{13}$$^{fl/fl}$ mice were gifts obtained from Dr. Stefan Offermanns' lab (Max Planck Institute for Heart and Lung Research, Bad Nauheim, Germany). PF4-Cre mice, LysM-Cre (Lyz2-Cre), and C57BL/6 mice were obtained from the Jackson Laboratory. Platelet-specific or leukocyte-specific Gα$_{13}$ knockout mice were generated by breeding Gα$_{13}$$^{fl/fl}$ and PF4-Cre or LysM-Cre mice and confirmed by genotyping and western blot analysis (Supplementary Fig. 7). Control mice were Cre only without the Gα$_{13}$ floxed allele or negative for Cre recombinase with matched genetic background, age, and sex. Animals were housed and bred in the Biologic Resources Laboratory at the University of Illinois at Chicago under 12 h light-dark cycles, controlled temperature (~23 degrees) and 40–50% humidity with free access to food and water. All animal procedures complied with the animal care and usage standards set forth by the National Institutes of Health and were approved by the Institutional Animal Care Committee, University of Illinois at Chicago (animal protocol number: 19-215). A randomized approach of choosing mice was used throughout the study, using all mice with the correct genotype without bias.

**Reagents**. Mouse anti-integrin β$_2$ antibody (1.BB.246, sc-71397, 2 µg per 500 µg lysate) and rat anti-integrin α$_{IIb}$ antibody (MWReg30, sc-19963, 1:250 for IHC staining) were purchased from Santa Cruz Biotechnology. Rabbit anti-integrin β$_2$ monoclonal antibody (D4N5Z, #73663, 1:1000 for western blot) was purchased from Cell Signaling Technology. Mouse anti-integrin β$_3$ monoclonal antibody M15 (2 µg per 500 µg lysate) was a gift from Dr. Mark Ginsberg (UCSD, CA). Rabbit anti-integrin β$_3$ antibody (18309-1-AP, 1:1000 for western blot) was obtained from Proteintech. Rabbit anti-Gα13 antibody (GTX32613, 1:1000 for western blot) was purchased from GeneTex. Rabbit anti-fibrin/fibrinogen polyclonal antibody (A0080, 1:2000 for IHC staining) was obtained from Dako/Agilent. FITC-conjugated rat anti-mouse P-selectin antibody was purchased from BD Pharmingen (553744). Mouse cytokines IL-6, TNFα, and IL-10 ELISA kits were obtained from R&D Systems (Minneapolis, MN). The DetectX® Urea Nitrogen (BUN) Detection Kit was purchased from Arbor Assays (Ann Arbor, MI). Creatinine and alanine transaminase colorimetric assay kits were from Cayman Chemical (Ann Arbor, MI). Mouse Cystatin C ELISA kit was from ThermoFisher Scientific (Waltham, MA). Mouse thrombin-antithrombin complexes ELISA kit (TAT, ab137994) and mouse fibrinogen ELISA kit (ab213478) were obtained from abcam (Cambridge, MA). Mouse D-Dimer (D2D) ELISA kit (EKC36716) was purchased from Biomatik (Wilmington, DE). Rabbit anti-VWF antibody (AB7356) and Poly (2-hydroxyethyl methacrylate) polymer (192066) was purchased from Millipore-Sigma (Kankakee, IL). PE/Cyanine7 labeled rat anti-mouse Ly-6G Antibody (clone 1A8, 127618) was purchased from Biolegend (San Diego, CA).

**HLPN preparation**. MB2mP6 (Myr-FEKEKL) and scrambled control (Myr-EFK-KLE) peptides were synthesized and purified by the Research Resources Center at the University of Illinois at Chicago or custom-made by CPC Scientific (San Jose, CA) in some experiments. PEG2000-DSPE (Avanti Polar Lipids Inc. Alabaster, AL), L-α- phosphatidylcholine (egg PC, Type XI-E, Sigma-Aldrich, St. Louis, MO) and peptides were mixed at a molar ratio of 55.6:11.9:40. HLPN were prepared using a film rehydration method[23,39].

**Cell culture and differentiation of THP-1 cells**. THP-1 cells (ATCC TIB-202) were maintained in RPMI-1640 supplemented with fetal bovine serum (10%); 2-mercaptoethanol (0.05 mM); HEPES (10 mM) and sodium pyruvate (1 mM). Cells were grown to a density of 6–8 × 10$^5$ cells mL$^{-1}$ and subjected to subculturing or used for experiments at no more than passage number 10. For differentiation to a macrophage phenotype[40], THP-1 cells were incubated with 100 nM phorbol 12-myristate 13-acetate (PMA, P8139, Millipore Sigma) at 2 × 10$^5$ cells mL$^{-1}$. After 24 h incubation, PMA-containing RPMI-160 medium was replaced with fresh medium, and cells were rested for 48 h before subsequent experiments. To keep the differentiated THP-1 cells in suspension conditions, we used 2 mg mL$^{-1}$ poly (2-hydroxyethyl methacrylate) polymer (poly-HEMA, in 95% ethanol) precoated plates to prevent cell adhesion during culturing[41,42].

**Co-immunoprecipitation**. Co-immunoprecipitation of integrin β$_2$ or β$_3$ and Gα$_{13}$ was performed similarly to the previously described procedure[23,24,43]. Briefly, differentiated THP-1 cells (2 × 10$^6$) or human platelets (5 × 10$^8$ mL$^{-1}$, 300 µL) were treated with MB2mP6 HLPN (50 µM) or scrambled peptide HLPN (50 µM). Ten min after treatment, THP-1 cells or platelets were stimulated with LPS (100 ng mL$^{-1}$) or thrombin (0.025 U mL$^{-1}$) for various time points and solubilized with NP40 lysis buffer (50 mM Tris, pH 7.4, 10 mM MgCl$_2$, 150 mM NaCl, 1% NP-40, 1 mM EGTA, 1 mM sodium orthovanadate, 1 mM NaF) with complete protease inhibitor cocktail tablets (Roche). After centrifugation at 17,800 × g for 10 min at 4 ºC, lysates were then collected and immunoprecipitated with mouse anti-integrin β$_2$ antibody (1.BB.246, 2 µg per 500 µg lysate) or mouse anti-integrin β$_3$ IgG (M15, 2 µg per 500 µg lysate) and an equal amount of mouse IgG overnight at 4 ºC, and then with protein A/G plus agarose beads (sc-2003, Santa Cruz Biotechnology, Inc, Dallas, TX) for 1 h at 4 ºC. Following three washes with NP40 lysis buffer, immunoprecipitants were analyzed by western blot.

**Detection of cytokine expression in THP-1 cells and mouse macrophages.** PMA-differentiated THP-1 cells were collected using 10 mM EDTA in PBS and seeded onto six-well plates for 14 h. After serum starving for 4 h, the THP-1 cells were stimulated with LPS for 8 h. The total RNA was purified from the THP-1 samples and quantitative RT-PCR was performed to detect different human cytokine expression using specific human cytokine primers (Supplementary Table 1). Mouse BMDMs were isolated from C57BL/6 mice[44]. After differentiating into macrophages by incubating in DMEM containing 15% L929-conditioned medium and 10% FBS for 7 days, BMDMs ($2 \times 10^6$ mL$^{-1}$) were seeded into six-well plates for 14 h and serum starved for 4 h. Eight hours after LPS stimulation, mouse BMDMs were collected and subjected to total RNA extraction and quantitative RT-PCR (qRT-PCR) analysis using SYBR Green (Roche) with various mouse cytokine-specific primers.

**THP-1 cell adhesion assay.** To test the adhesion of differentiated THP-1 cells on human ICAM-1 surface, we precoated the 48-well plates with recombinant hICAM-1 (10 µg mL$^{-1}$, Fc Chimera Protein from R&D Systems, 720-IC) for 1 h inside a 37 °C incubator and washed two times with PBS before using. Differentiated THP-1 cells ($5 \times 10^5$ mL$^{-1}$) were pretreated with various concentrations of MB2mP6 or scrambled peptide HLPN in RPMI 1640 cell culture medium for 20 min and loaded into ICAM-1 precoated 48-well plates (200 µL). After allowing cells to adhere for 30 or 60 min, the cell medium was aspirated, and the non-adherent cells were washed away with PBS. The adherent THP-1 cells were stained with 0.5% crystal violet solution for 20 min at room temperature[45] and quantified by reading the plates at OD590nm on a SpectraMAX 340 plate reader using SoftMAX pro software (version 2.2.1) from Molecular Devices (San Jose, CA).

**Platelet preparation, aggregation, and granule secretion.** Human blood was drawn by venipuncture from healthy volunteers. Institutional Review Board approval was obtained from the University of Illinois at Chicago, and informed consent from volunteers was obtained in accordance with the Declaration of Helsinki.

To prepare platelet-rich plasma (PRP), whole blood was anticoagulated with 3.8% trisodium citrate. To prepare human platelets, one-seventh volume of ACD was used as anticoagulant. Platelets were washed twice and resuspended in modified Tyrode's buffer[46]. Platelet aggregation and adenosine triphosphate (ATP) secretion were measured simultaneously in a lumiaggregometer (Chronolog) using Aggro/Link8 (version 1.2.9) from CHRONO-LOG CORP at 37 °C with stirring (1000 rpm)[47]. For P-selectin expression assays, mouse platelets were used. Mouse platelets were isolated from mouse PRP and washed in CGS buffer (sodium chloride 0.12 M, D-glucose 0.03 M, trisodium citrate 0.0129 M, pH6.5) and resuspended in modified Tyrode buffer (12 mM NaHCO$_3$, 138 mM NaCl, 5.5 mM glucose, 2.9 mM KCl, 0.42 mM NaH2PO4, 10 mM N-2-hydroxyethylpiperazine-N'-2-ethanesulfonic acid, pH 7.4) containing 1 mM CaCl$_2$ and 2 mM MgCl$_2$[48]. Washed platelets in Tyrode buffer were preincubated with 20 µM MB2mP6 or scrambled peptide HLPN for 5 min, and then stimulated with various concentrations of thrombin at 37 °C for 5 min. After fixing with 2% paraformaldehyde, the platelets were incubated with FITC-conjugated rat anti-mouse P-selectin antibody (BD Pharmingen, 553744) for 30 min at room temperature. After dilution 10 times in PBS (with 1%BSA), P-selectin expression was analyzed using a Accuri C6 flow cytometer.

**Fibrinogen binding assay.** To measure fibrinogen binding, $3 \times 10^8$ mL$^{-1}$ washed mouse platelets in modified Tyrode's buffer containing 0.1% bovine serum albumin (BSA) were preincubated with scrambled peptide HLPN or MB2mP6 HLPN (20 µM) for 10 minutes. Oregon Green-conjugated fibrinogen (10 µg mL-1, Molecular Probes) was then added, and the platelets were stimulated with 500 µM PAR4AP. After 30 minutes, the reaction was diluted with PBS containing 1% BSA. Platelet-bound fibrinogen was detected by flow cytometry using an Accuri C6 flow cytometry with CFlow Plus software (version 1.0.227.4) (BD Biosciences). Gating strategy used for flow cytometry analysis is shown in Supplementary Fig. 9.

**Neutrophil phagocytosis and bacterial killing assays.** Mouse neutrophil phagocytosis was performed as previously described[49]. Mouse neutrophils were isolated from the bone marrow of C57 wild type mice[50] and resuspended in RPMI 1640 medium containing 10% FBS. After resting at 37 °C for 30 min, the neutrophils ($2 \times 10^6$ mL$^{-1}$, 2 mL) were seeded onto six-well plates and pretreated with MB2mP6 or scrambled peptide HLPN at 37 °C for 20 min. At the same time, Alexa Fluor 488 conjugate E. coli (K-12, Thermo Fisher Scientific) BioParticles were opsonized with 10% normal mouse serum in RPMI 1640 at 37 °C for 1 h. After three washes with RPMI 1640 containing 10% FBS, the E. coli BioParticles were added to a six-well plate preloaded with mouse neutrophils at a ratio of 10:1 (BioParticles: neutrophil) and incubated for 60 min. After fixing with 4% PFA solution for 15 min at room temperature, the adherent neutrophils were detached from the six-well plate and then quenched with Trypan Blue solution to exclude cell surface-bound BioParticles. The neutrophils were confirmed by staining with surface marker Ly6G using PE/Cyanine7 labeled rat anti-mouse Ly-6G Antibody (clone 1A8). The intake BioParticles were detected and analyzed by Accuri C6 flow cytometry using CFlow Plus software (version 1.0.227.4) from BD Biosciences. Neutrophil bactericidal activity was tested using fresh cultured E. coli bacteria

(K-12)[51]. In brief, the live E. coli were opsonized by incubating with 10% normal mouse serum in HBSS at 37 °C for 1 h and then mixed with mouse neutrophils (10:1) pretreated with MB2mP6 or scrambled peptide HLPN. At the indicated time point, the bacteria and neutrophil mixture (50 µL) was collected and lysed with 950 µL pH 11 H$_2$O. To determine the number of live bacteria in the samples, serial dilutions of the lysate were plated onto TSA agar with 5% sheep blood plates. The bacterial colonies were recorded 16–18 h later and counted by ImageJ software.

**FeCl$_3$-induced mouse carotid artery thrombosis model.** Eight- to 10-week-old C57BL/6 mice (20–25 g body weight) were anesthetized by isoflurane inhalation. The right carotid artery was isolated. A MA-0.5SB nanoprobe (Transonic Systems, Ithaca, NY) was hooked to the carotid artery to monitor blood flow with a TS420 flowmeter (Transonic Systems)[52]. After stabilization, a filter paper disc (2 mm diameter) soaked with 1.2 µL of 7.5% FeCl$_3$ (Sigma Aldrich, St. Louis, MO) was placed on top of the carotid artery for 3 min to induce thrombosis and then removed. MB2mP6 or scrambled control peptide were injected 15 min before the procedure retro-orbitally. Blood flow was monitored continuously until 5 min after occlusion or for 15 min if no occlusion occurred. Time to occlusion was defined as the time between the removal of the filter paper and stable occlusion (no blood flow for 5 min). Data were analyzed using the Mann–Whitney test to compare medians between treatments by Graphpad Prism software (version 8.3.0).

**Tail bleeding time.** Tail bleeding time was performed as previously described[23,52]. Eight- to 10-week-old C57BL/6 mice weighing 20–25 g were anesthetized with isoflurane. After cutting a 0.5-cm-long segment off the distal tip of the tail, their tails were immersed in 0.15 M NaCl immediately at 37 °C. Tail bleeding time was defined as the time between cutting the tail and stable cessation of bleeding (no re-bleeding within 60 s). Bleeding was observed for up to 15 min. If bleeding persisted at 15 min, bleeding was stopped by application of pressure. Data were analyzed using the Mann–Whitney test to compare medians between groups by Graphpad Prism software (version 8.3.0).

**rpA reaction.** C57/BL6 mice were anesthetized by i.p. injection of a mixture of ketamine and xylazine (100 mg kg$^{-1}$ and 16 mg kg$^{-1}$ mouse weight), and the medial surface of the mouse back was shaved. Shaved mice received intradermal injections of rabbit anti-BSA antibody (6 µg µL$^{-1}$, MP Biomedicals, OH) in 25 µL of PBS, followed by i.v. injection of BSA (75 µg g$^{-1}$ mouse weight) as well as the peptide inhibitors (5 µmol kg$^{-1}$ mouse weight) in 100 µL sterile 0.9% NaCl. In addition, control sites received 25 µL of PBS in parallel. Four hours after injection, mice were euthanized and inflamed or control skin samples at each injection site were collected. The Hb contents in the skin samples were quantified by Hb colorimetric assay kit (Cayman Chemical).

**CLP sepsis model.** CLP sepsis was induced as described previously[53,54]. In brief, mice (14- to 16-week-old with an equal number of each gender) were anesthetized by intraperitoneal administration of ketamine (100 mg kg$^{-1}$ body weight) and xylazine (8 mg kg$^{-1}$). After a midline laparotomy, the cecum was ligated at about 1 cm from the end and then subjected to a double "through and through" perforation with an 18-gauge needle. Sham-operated mice underwent the same procedure except for ligation and puncture of the cecum. After the procedure, an analgesic (buprenorphine, 0.1 mg kg$^{-1}$, s.c.) was given immediately and every 12 h thereafter for 3 d. At these time points, fluid resuscitation (prewarmed 0.9% NaCl, 0.05 mL g$^{-1}$ body weight) was given through subcutaneous injection to prevent fluid loss and help the recovery of body temperature. Antibiotic (Claforan® solution; 10 mg kg$^{-1}$ body weight) was subcutaneous administrated (to mimic clinical conditions) at the time MB2mP6 treatment was initiated and lasted for 5 days. The medication was continued until the situation of scheduled sacrifice, death, or study completion in the surviving mice. The survival of mice was observed every 6 h for 8 days and analyzed by the Log Rank test (GraphPad Prism software, San Diego, CA). For CLP-induced mouse organ injury, blood samples and kidney or lung tissues were collected 24 h after CLP and subjected to cytokine and immunohistochemistry analyses.

**Mouse jugular vein cannulation and continual peptide infusion.** Mice were weighed, anesthetized using ketamine/xylazine (100/16 mg kg$^{-1}$), and transferred to a heated platform under a dissection microscope. A 5 mm incision through the skin on the upper part of the mouse back was made posterior to the ears and between the scapulae. The mouse was then placed in a supine position and the skin over the right-side jugular vein was shaved and cleaned with hexachlorophene and 70% ethanol. After a vertical incision over the jugular site, the vein was dissected and exposed. The catheter was tunneled from the upper back incision to the jugular incision through a trocar sleeve kit. The jugular vein was carefully separated, and two sutures were drawn around the vessel and tied loosely. Then, a small incision was made in the vein, the beveled catheter was inserted and tied in place with a suture, without closing off the catheter. The wound was closed using a 6-0 suture and cleaned with hexachlorophene. For peptide treatment, MB2mP6 or control peptide micelles were injected in a bolus dose of 2.5 µmol kg$^{-1}$ followed by a continuous infusion at a rate of 1.25 µmol kg$^{-1}$ h$^{-1}$ for 5 days.

**Measurement of organ injury.** Twenty-four hours after CLP, mouse blood was collected and added to an EDTA-rinsed microcentrifuge tube or citrate buffer. The plasma was obtained after centrifugation at $1000 \times g$ at 4 ºC for 5 min. In some cases, mouse serum was collected from blood without adding any anti-coagulant. The kidney function markers BUN, creatinine and cystatin C and mouse coagulant factors fibrinogen and TAT were detected using corresponding kits according to the manufacturers' instructions. Septic mouse kidney, lung, and liver tissue were collected from the sacrificed mice 24 h after CLP, as well as from sham-operated mice and fixed in 10% formalin.

**Immunohistochemistry.** Mouse kidney and lung were collected 24 h after CLP after perfusion with phosphate-buffered saline and then fixed in 10% formalin solution. Twenty hours after fixation, mouse tissues were dehydrated in 70% ethyl alcohol and embedded in paraffin. Four to five-micrometer sections were cut and stained with hematoxylin and eosin for quality control. Sections were de-paraffinized, rehydrated, peroxidase blocked, and subjected to antigen retrieval by heating at 95 ºC for 20 min in pH 6.0 citrate buffer. Rat anti mouse integrin αIIb antibody (1:250, Clone MWReg 30), rabbit anti fibrin/fibrinogen antibody (1:2000, A0080, Dako), rabbit anti-VWF antibody (1:50, AB7356, Millipore Sigma) and anti-rat or anti-rabbit IgG-avidin-biotin complex kits (Vector Laboratories) were used to stain platelet-rich or fibrin-rich thrombi. The positive stain in each section was quantified with ImageJ software. Slides were also stained with Mallory's PTAH (PTAH Stain kit, American MasterTech, McKinney, TX) to identify fibrin deposition and viewed with a Leica DMI RB microscope using a $40 \times /0.55$ NA objective. Total thrombotic area/glomerulus was quantitated by analyzing 20–30 glomeruli from each group using ImageJ software.

**Detection of cytokine expression.** Twenty-four hours after CLP, mouse blood was collected from sham-operated and CLP-induced septic mice. The serum was isolated and analyzed for cytokine levels using specific mouse cytokine ELISA kits. Mouse lungs were also collected from those sacrificed mice after PBS perfusion. Cytokine transcripts expressed in lung tissue were detected by real-time PCR using SYBR green method.

**Mouse lung microvascular permeability assay.** Mouse lung microvascular permeability was determined as described previously[54]. Briefly, mice were i.v. injected with Evans blue albumin (EBA, Sigma, $25\,\text{mg}\,\text{kg}^{-1}$) 30 min before the termination of CLP-induced septic mice to assess vascular leakage. After 30 min, the lungs were perfused with PBS and excised out of the thoracic cavity. After homogenization in PBS (1 mL per 100 mg of lung tissue), lungs were further incubated in two volumes of formamide (Sigma) for 18 h at 65 °C. At the end of this incubation, the homogenate was centrifuged at $17,800 \times g$ for 30 min and the supernatant was used to determine optical density spectrophotometrically at 620 nm. A standard curve was plotted, and EBA concentration in each sample was calculated as micrograms of Evans blue present in each milligram of lung tissue.

**Detection of fecal occult blood in septic mice.** Semi-quantitative fecal occult blood test was performed as previously described[55]. In brief, mouse feces were harvested before and 24 h after CLP surgery. The feces were weighed and mixed with $ddH_2O$ at 1 mg mL$^{-1}$ in a microtube. After vortexing the tube for 2 min, the mixtures were centrifuged for 3 min at $8000 \times g$. Five microliters of supernatant was transferred to a new microtube and mixed with 100 μL of fresh prepared 0.04% luminol solution containing 0.53 M potassium hydroxide and 1.5% hydrogen peroxide. After mixing, the chemiluminescence signal in the sample was measured immediately by a luminometer (Femtomaster FB 12, Zylux Corporation).

**Statistical analysis.** Data are expressed as means ± SEM. For parametric data, differences between groups of samples were evaluated with student's $t$-test, one-way ANOVA, or two-way ANOVA with GraphPad Prism software. For non-parametric data, statistical significance was determined using the Mann–Whitney test. Survival analysis was performed using Graphpad Prism software with the Kaplan–Meier method. A $p$-value $\leq 0.05$ was considered statistically significant.

**Reporting summary.** Further information on research design is available in the Nature Research Reporting Summary linked to this article.

## Data availability

The authors declare that the data supporting the findings in the current study are available in the Article and its Supplementary Information files. Additional information can be obtained from the corresponding author upon reasonable request. Source data are provided with this paper.

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

## Acknowledgements
We thank Dr. Stefan Offermanns for supplying the Gα13 floxed mice. This work is partially supported by National Heart, Lung and Blood Institute grants R35HL150797 (X. D.), RO1HL125356 (X.D.), RO1HL080264 (X.D.), RO1HL062350 (X.D.), R43 HL142396 (M.K.D. and R.A.S.), and Vascular Interventions/Innovations and Therapeutic Advances (VITA) Stage-A (HHSN268201400007C) (X.D.) and Stage B (HHSN268201700002C) (Dupage Medical Technology) contracts.

## Author contributions
N.C. contributed to the performance of experiments, data analysis/interpretation, and manuscript preparation; Y.Z., M.K.D., Y.B., and C.W. performed some experiments and analyzed data; R.A.S. prepared and revised the manuscript; X.D. designed research, analyzed/interpreted data, and wrote the manuscript.

## Competing interests
Patents related to this study: U.S. Provisional Application No. 62/932,024 (University of Illinois), X.D. has ownership interests in DMT Inc., which licenses the university patents. R.A.S. is employed by DMT Inc. The other authors declare no competing interests.
