## [Peer Review File · Nature Communications]

Reviewers' comments:

Reviewer #1 (Remarks to the Author):

Citing the literature, the authors conclude that in sepsis in humans, there was significant hemorrhage with Xigris. However, this was largely seen in septic pediatric patients who developed intracranial hemorrhage, leading to the FDA prohibiting the use of Xigris in pediatric cases with sepsis. In adults, the withdrawal (after approval) of Xigris was due to lack of efficacy in septic adult humans, not because of hemorrhagic complications, as determined by a third clinical trial.

The authors describe a drug, MB2mP6 that targets activation of $\beta 2$ and $\beta 3$ integrins of platelets and leukocytes, resulting in increased survival of mice with polymicrobial sepsis. These animals, when septic, do not usually demonstrate significant signs of hemorrhagic complications, although analysis of blood shows evidence of consumptive reduction of clotting factors. The authors indicate that in mice their drug reduces "systemic inflammation" and absence of hemorrhagic complications. The authors clearly show after CLP that survival improved and cytokine levels are negatively affected. They also show that the reduced levels of fibrin in septic mice are protected in K.O. mice. Other outcomes in septic mice treated with the inhibitor, such as multiorgan dysfunction are also described, although the results in treated mice are marginal at best.

Because the authors focus on the issue of hemorrhaging during inflammation, they should carry out quantitative measurements of clotting factors in their experimental model.

Reviewer #2 (Remarks to the Author):

Cheng et al

The study by Cheng et al. presents suggests a new concept for treating systemic inflammation using a drug targeting both inflammation and thrombosis. Specifically, the authors propose to use a peptide inhibitor to inhibit outside-in signaling (Integrin-Ga13 interaction) in leukocytes and platelets. Inhibitor studies are complemented by studies with mice deficient in Ga13 in leukocytes and/or platelets. Thromboinflammation is induced by cecal ligation and puncture (CLP). The paper is well-written and the presented results for the CLP model are impressive. However, there are some concerns that need to be addressed.

Major:

1. use of Cre-negative control mice. The use of Cre-negative mice as controls is a bad choice as Cre expression itself (in the absence of any floxed genes) can affect biology - see statement on the Jackson laboratories page (<https://www.jax.org/news-and-insights/jax-blog/2013/september/a-dozen-facts-you-didnt-know-about-cre-lox>). Key experiments with LysM/PF4-Cre mice have to be repeated with Cre-positive controls in order to rule out Cre artifacts.
2. Specificity of the peptide inhibitor. What studies were performed to show that the MB2mP6 peptide specifically affects outside-in signaling? A more careful characterization of leukocyte and platelet functions in the presence of the inhibitor is necessary. How is it explained that MB2mP6 inhibits LPS-induced cytokine expression in THP-1 cells - what is the contribution of integrin outside-in signaling in this context?
3. Usefulness of MB2mP6-like drug in a clinical setting. The authors show that giving the peptide 6 hours after CLP-induction has a mild protective effect. Given that this time point is early in the course of CLP one would not expect the drug to be helpful in patients showing up in the hospital where disease will be further along. Have studies been done in mice at later time points after CLP induction? How long can a protective effect be seen?
4. Another weakness of the study is that the inhibitory activity of the peptide was not characterized ex vivo. There are effects in the CLP model, but the extent of inhibition is not reported. Why were co-IP studies not performed?

Minor:

1. Fibrin and platelet accumulation: why was that measured as %area rather than an absolute value? Hard to determine if platelet adhesion and fibrin formation were really reduced. Also, why was neutrophil infiltration not reported.
2. Why is new peptide less potent than M3mP6 peptide in the FeCl3 carotid artery model. Where co-IP and platelet aggregation studies done side-by-side?

Reviewer #3 (Remarks to the Author):

This is an excellent manuscript that convincingly shows that Ga13 in platelets and myeloid cells makes distinct contributions to sepsis associated pathology that probably have added effects on sepsis mortality. These data, based on findings from mice lacking ga13 in either cell type alone or in both cell types are solid and important. The authors also show that a cell permeable peptide that partially blocks the association of the platelet integrin, $\alpha 11\text{bb}3$ and the leukocyte integrin, $\alpha 2\text{bm}$, with Ga13 and inhibits platelet aggregation and LPS-induced increases in pro-inflammatory cytokine gene expression in THP1 cells. In vivo administration of this peptide is quite effective in improving mortality in the CLP model of sepsis and in reducing intravascular coagulation, renal injury and increased circulating cytokine levels without increasing bleeding time. Administration of the same peptide 6 hours after CLP also improves survival, but to a much lesser degree. No data are included about cytokine levels or AKI in mice treated at 6 h. Although the authors cite other studies suggesting mice are quite ill at 6 hours after CLP, they provide no evidence that this was the case in their hands, and mortality was not seen until 24 hours. Overall, this is important work that does suggest it could be possible to therapeutically target Ga13 activation by integrins to inhibit critical but distinct effects of integrin signaling in platelets and leukocytes that together contribute to sepsis-induced pathology. My only significant concern is that the authors unnecessarily oversell their evidence that the peptide they have designed might be therapeutically useful. Most patients, like most mice in my experience, are not obviously ill within the first few hours after the insult that results in sepsis, so an effective therapeutic would likely need to have a potent effect when administered much later, ideally at a point when the mice were clearly quite ill (probably more like 18-24 hours after CLP). I would simply suggest that the authors be more circumspect about the likelihood that they have designed a cure for sepsis and focus more on the impressive and interesting biology.

Minor comment-

In Figure 1 f the y axis is labeled ATP but the legend says ADP.

Dean Sheppard

Reviewer #4 (Remarks to the Author):

This is an interesting manuscript on an important subject. It recognizes the fact that, most often, platelet and innate immune function are activated together and share molecular players. The writing, especially the introduction, is skimpy and does not explain the concept of inside-out and outside-in integrin signaling well. Especially, in the thrombin activation. A general reader will not know that here an extremely low concentration of thrombin was used to only partially activate the integrin so that ligand binding and, thus, induction of outside-in signaling, was necessary for full platelet activation. While the experimental work presented is highly relevant to the subject at hand, the scrambled peptide should always be used as a control. From the writing, it was not clear how the peptide gets in the cells. Was it myristoylated, as the name implies MB2mP6 (Myr-FEKEKL) and in micelles, as indicated by the methods, in all experiments? How do the micelles get to the integrin tails? Their proximity to plasma membrane? This is not well explained, although it must be known. In animal experiments, does MB2mP6 inhibit $\beta 3$ integrin also in endothelial cells?

Specific comments:

1. Neutrophil functionality. While neutrophil activity can be detrimental in inflammation, it is also lifesaving in infection. From the CLP studies, the drug does not seem to have major detrimental effects, however, this should be demonstrated more convincingly both in vitro and in vivo. The effect of MB2mP6 on phagocytosis of bacteria in vitro and on the bacterial count (bacteremia) in the CLP sepsis model needs to be documented.
2. In vitro platelet work. If effects of MB2mP6 on platelets were not previously published, they should be shown in greater detail, i.e. representative tracing is not sufficient (several preparations and statistics should be used). Other agonists, such as ADP and high thrombin concentration, should be shown to demonstrate that inside-out signaling was not inhibited. Is alpha-granule secretion affected by the drug; P-selectin expression?
3. Bleeding in animals treated with MB2mP6. Should be examined also in the stool/intestine contents, as was done for $\beta 3$ -/- mice characterization. Hemorrhaging in CLP mice should be evaluated by histology in other organs other than kidney, for example, in the lung.
4. Fibrin deposition. Does not represent thrombosis unless platelets are also in evidence. Fibrin alone may indicate vascular leakage. Platelet staining and, perhaps, VWF staining in addition to fibrinogen would make the presence of thrombosis more convincing.

Point-by-point response to review comments:

Reviewer #1 (Remarks to the Author):

Citing the literature, the authors conclude that in sepsis in humans, there was significant hemorrhage with Xigris. However, this was largely seen in septic pediatric patients who developed intracranial hemorrhage, leading to the FDA prohibiting the use of Xigris in pediatric cases with sepsis. In adults, the withdrawal (after approval) of Xigris was due to lack of efficacy in septic adult humans, not because of hemorrhagic complications, as determined by a third clinical trial.

Response: From clinical trial results, it appears that the first trial that led to FDA approval demonstrated significant efficacy but was also associated with significant bleeding. The repeat PROWESS-SHOCK trial failed to replicate the beneficial effects of Xigris in the initial PROWESS trial, but the reason for the discrepancy was never proved. However, the repeat trial observed unusually low major bleeding events, leading to the suggestion that a less effective drug was used in the repeat trial. Several postmarket studies clearly showed significant adverse effect of hemorrhage, and it was reported that the adverse effect of hemorrhage outweighed the benefit of this drug, especially in patients with bleeding tendencies. While we do not try to make conclusions out of these previous reports, there is no mechanistic reason (nor data) to believe that an efficacious Xigris is not associated with increased hemorrhage. To address the reviewer's comment, we have now rewritten the description of previous clinical trials to be more accurate.

The authors describe a drug, MB2mP6 that targets activation of β_2 and β_3 integrins of platelets and leukocytes, resulting in increased survival of mice with polymicrobial sepsis. These animals, when septic, do not usually demonstrate significant signs of hemorrhagic complications, although analysis of blood shows evidence of consumptive reduction of clotting factors. The authors indicate that in mice their drug reduces "systemic inflammation" and absence of hemorrhagic complications. The authors clearly show after CLP that survival improved and cytokine levels are negatively affected. They also show that the reduced levels of fibrin in septic mice are protected in K.O. mice. Other outcomes in septic mice treated with the inhibitor, such as multiorgan dysfunction are also described, although the results are in treated mice are marginal at best.

Because the authors focus on the issue of hemorrhaging during inflammation, they should carry out quantitative measurements of clotting factors in their experimental model.

Response: We thank the reviewer for the suggestion. We have performed new experiments (Supplementary Figs. 2 and 5) to show that fibrinogen and thrombin-antithrombin complex (TAT) levels in the blood were increased as early as 6 hours after CLP. The TAT complex in the septic mice treated with MB2mP6 was significantly decreased (Supplementary Fig. 5), suggesting that whereas MB2mP6 did not significantly affect the levels of fibrinogen in septic mouse blood, the activation of coagulation system during CLP-induced sepsis as indicated by TAT was inhibited by MB2mP6.

Reviewer #2 (Remarks to the Author):

The study by Cheng et al. presents suggests a new concept for treating systemic inflammation using a drug targeting both inflammation and thrombosis. Specifically, the authors propose to use a peptide inhibitor to inhibit outside-in signaling (integrin-Ga13 interaction) in leukocytes and platelets. Inhibitor studies are complemented by studies with mice deficient in Ga13 in leukocytes and/or platelets.

Thromboinflammation is induced by cecal ligation and puncture (CLP). The paper is well-written and the presented results for the CLP model are impressive. However, there are some concerns that need to be addressed.

Response: We thank the reviewer for the encouraging comments.

Major:

1. use of Cre-negative control mice. The use of Cre-negative mice as controls is a bad choice as Cre expression itself (in the absence of any floxed genes) can affect biology - see statement on the Jackson laboratories page (<https://www.jax.org/news-and-insights/jax-blog/2013/september/a-dozen-facts-you-didnt-know-about-cre-lox>). Key experiments with LysM/PF4-Cre mice have to be repeated with Cre-positive controls in order to rule out Cre artifacts.

Response: We thank the reviewer for the suggestion to use Cre mouse controls in studying the effect of $G\alpha_{13}$ knockout in mouse sepsis survival probability. Breeding these mice and repeating these experiments are a major reason for the delay in resubmission of this revised manuscript during the pandemic. As suggested, Cre mice are now used as controls in all these experiments. Overall, we found that LysM-cre mice and PF4-cre mice are not significantly different from the $G\alpha_{13}^{fl/fl}$ mice in survival probability. In contrast, $G\alpha_{13}^{fl/fl/LysMcre}$ and $G\alpha_{13}^{fl/fl/PF4cre}$ showed significantly increased survival rate compared to their respective fl/fl controls and cre controls. Similarly, $G\alpha_{13}^{fl/fl-LysMcre/PF4cre}$ dual knockouts also showed significantly increased survival rate as compared with either LysM-cre/PF4-cre control mice or fl/fl mice. The overall conclusions of the in vivo sepsis mouse models are now re-confirmed using two different sets of controls.

2. Specificity of the peptide inhibitor. What studies were performed to show that the MB2mP6 peptide specifically affects outside-in signaling? A more careful characterization of leukocyte and platelet functions in the presence of the inhibitor is necessary. How is it explained that MB2mP6 inhibits LPS-induced cytokine expression in THP-1 cells - what is the contribution of integrin outside-in signaling in this context?

Response: We thank the reviewer for the constructive comments. We have performed additional experiments to address the reviewer comments: (1) MB2mP6 did not affect PAR4AP-induced fibrinogen binding to platelets and did not affect primary platelet aggregation-induced by ADP or high dose thrombin (Fig. 3), indicating that MB2mP6 did not affect inside-out signaling and the ligand binding function of platelet integrin $\alpha_{IIb}\beta_3$ in platelets. (2) MB2mP6 selectively inhibited low-dose thrombin-induced platelet granule secretion during platelet aggregation, which is known to be integrin-dependent, and secretion-dependent amplification of platelet aggregation, which is a function of integrin-outside-in signaling, indicating that MB2mP6 inhibits outside-in signaling. (3) Similarly in THP-1-derived macrophages, MB2mP6 did not affect cell adhesion, suggesting that integrin activation and the ligand binding function of integrins in macrophages are not affected (Fig. 1). However, MB2mP6 potently inhibited adhesion-dependent cytokine expression in these macrophage-like cells (Fig. 1), suggesting that MB2mP6's inhibition of cytokine expression is outside-in signaling-dependent. (4) MB2mP6 inhibited $G\alpha_{13}$ -integrin interaction (Fig. 1a and 3a) which was previously shown to selectively inhibit outside-in signaling. Additionally, in a separate on-going project on the mechanism of neutrophil migration, we have obtained evidence that MB2mP6 inhibited β_2 -dependent neutrophil spreading without affecting neutrophil adhesion to β_2 ligand ICAM1. Together, these data suggest that the inhibitor selectively inhibits $G\alpha_{13}$ -dependent early phase integrin outside-in signaling.

Figure for reviewer viewing only: MB2mP6 does not affect neutrophil adhesion on ICAM1 (A) but inhibited neutrophil spreading on ICAM1 (B).

3. Usefulness of MB2mP6-like drug in a clinical setting. The authors show that giving the peptide 6 hours after CLP-induction has a mild protective effect. Given that this time point is early in the course of CLP one would not expect the drug to be helpful in patients showing up in the hospital where disease will be further along. Have studies been done in mice at later time points after CLP induction? How long can a protective effect be seen?

Response: To address this question, we further postponed the start point of peptide treatment to 18 hours after CLP onset in the mouse sepsis model. As shown in Fig 5c, MB2mP6 significantly improved survival probability of septic mice even when treatment started at 18 hours after CLP. It is important to note that, to mimic clinical situations, the antibiotic treatment was given at the same time as MB2mP6 (or control peptide) treatment. Thus, both MB2mP6 group and control group showed increased mortality rate when treatment was delayed to 18 hours. Also, in this severe mouse CLP model, sepsis is well established within 4-6 hours as evidenced by progressively or significant increases in cytokines (supplementary Fig. 2), and lung neutrophil recruitment, alveolar leak, endothelial damage, liver neutrophil and platelet recruitment with impaired sinusoidal perfusion, and acute kidney injury (1-5). Thus our data suggest that MB2mP6 not only prevents sepsis, but is also an effective treatment after the onset of sepsis in animal models that mimic typical clinical situations.

4. Another weakness of the study is that the inhibitory activity of the peptide was not characterized *ex vivo*. There are affects in the CLP model, but the extent of inhibition is not reported. Why were co-IP studies not performed?

Response: We showed that MB2mP6 inhibited $G\alpha_{13}$ and β_3 -integrin binding in platelets (co-IP assay in Fig. 3a) and $G\alpha_{13}$ and β_2 -integrin binding in THP-1 cells (co-IP assay in Fig. 1a) *in vitro*. We also showed that MB2mP6 inhibited cytokine secretion from bone marrow-derived macrophages *in vitro* (Supplementary Fig. 1). Additionally, our *in vitro* data demonstrated that MB2mP6 inhibited platelet secretion and secretion-dependent platelet aggregation (Fig. 3b-e). The newly added experiments further

demonstrate that MB2mP6 did not affect fibrinogen binding to platelets (Fig. 3f and 3g) as well as ADP-induced platelet aggregation *in vitro* (Fig. 3h). These were all *ex vivo* experiments. We also tested the effects of MB2mP6 on cytokine secretion, (Fig. 5d, 5e and supplementary Fig. 3), renal function (Fig. 3f and 3g), kidney microvascular thrombosis (Fig. 5f, 5g and supplementary Fig. 4a-c) and lung vascular leakage (Fig. 6) in septic mice.

Minor:

1. *Fibrin and platelet accumulation: why was that measured as %area rather than an absolute value? Hard to determine if platelet adhesion and fibrin formation were really reduced. Also, why was neutrophil infiltration not reported.*

Response: Since we detected fibrin and platelet accumulation by an immunohistochemical staining method, it's hard to control all reactions in the same slide (and different slides) in a linear range for quantification of an absolute value of reaction levels based on different paraffin slides. Thus, we determined the extent of thrombosis by quantifying the positive stain area to indicate changes of fibrin and platelets in septic tissues. We believe that this is a valid and important quantification, reflecting how extensive thrombosis occurs in the microvasculature in tissue. Neutrophil infiltration has unique and complex mechanisms, and thus requires an independent project to clearly elucidate, which is beyond the scope of the current manuscript. However, in a separate and on-going project, our lab has tested and demonstrated the effect of MB2mP6 in β_2 -integrin-dependent neutrophil trans-endothelial migration *in vitro* and *in vivo*.

2. *Why is new peptide less potent than M3mP6 peptide in the FeCl3 carotid artery model. Where co-IP and platelet aggregation studies done side-by-side?*

Response: This is a very good question, which we are very interested in studying. One *in vitro* difference may explain the *in vivo* effect: MB2mP6 became ineffective in inhibiting platelet ATP secretion at higher thrombin concentrations (>0.05 U/m) (Fig 3e), but M3mP6 is still effective at higher thrombin concentration in inhibiting ATP secretion (ref. 6 below). Structurally, a key difference between β_2 and β_3 $G\alpha_{13}$ binding sites is that β_2 has the EKE sequence, while β_3 has EEE sequence, a charge reversal, which may affect its function in these two different cells. However, due to limited time and resources and a large number of new experiments to be completed, we have to leave this interesting question to future studies as the mechanism of this difference does not affect the overall conclusion of our paper as much as other experiments.

Reviewer #3 (Remarks to the Author):

This is an excellent manuscript that convincingly shows that Gal3 in platelets and myeloid cells makes distinct contributions to sepsis associated pathology that probably have added effects on sepsis mortality. These data, based on findings from mice lacking Gal3 in either cell type alone or in both cell types are solid and important. The authors also show that a cell permeable peptide that partially blocks the association of the platelet integrin, $\alpha 11b\beta 3$ and the leukocyte integrin, $\alpha 2bm$, with Gal3 and inhibits platelet aggregation and LPS-induced increases in pro-inflammatory cytokine gene expression in THP1 cells. In vivo administration of this peptide is quite effective in improving mortality in the CLP model of sepsis and in reducing intravascular coagulation, renal injury and increased circulating cytokine levels without increasing bleeding time. Administration of the same peptide 6 hours after CLP also improves

survival, but to a much lesser degree. No data are included about cytokine levels or AKI in mice treated at 6 h. Although the authors cite other studies suggesting mice are quite ill at 6 hours after CLP, they provide no evidence that this was the case in their hands, and mortality was not seen until 24 hours. Overall, this is important work that does suggest it could be possible to therapeutically target $G\alpha_{13}$ activation by integrins to inhibit critical but distinct effects of integrin signaling in platelets and leukocytes that together contribute to sepsis-induced pathology. My only significant concern is that the authors unnecessarily oversell their evidence that the peptide they have designed might be therapeutically useful. Most patients, like most mice in my experience, are not obviously ill within the first few hours after the insult that results in sepsis, so an effective therapeutic would likely need to have a potent effect when administered much later, ideally at a point when the mice were clearly quite ill (probably more like 18-24 hours after CLP). I would simply suggest that the authors be more circumspect about the likelihood that they have designed a cure for sepsis and focus more on the impressive and interesting biology.

Response: We are very grateful for the reviewer's insightful comments. To address the reviewer's comments, we performed new experiments to test the effect of administration of MB2mP6 18 hours after onset of CLP on sepsis survival in mice (Fig. 5c). MB2mP6 showed significantly improved survival probability as compared with control peptide even when used at this late stage of sepsis. As discussed in addressing reviewer 2's question, delayed treatment of CLP mice to 18 hours resulted in increased mortality in both MB2mP6 group and control group. This is likely due to the fact that use of antibiotics which was included in the treatment protocol was also delayed. Thus, we believe that simultaneous targeting platelet and leukocyte $G\alpha_{13}$ with MB2mP6 has the potential for future development for sepsis treatment, based on the mouse model. However, we appreciate the reviewer suggestion to be cautious and to emphasize biology and revised the manuscript accordingly. To address the reviewer's comments with regard to the cytokine levels of mice at different times of sepsis in our experimental model, we tested the whole blood cytokines and coagulation factors in mice 6 hours and 18 hours after CLP as well as in the control mice with sham (Supplementary Fig. 2). We found that the IL-6 and TNF α levels were both significantly increased at 6 hours. The IL-6 level was decreased at 18 hours, although the late phase cytokine TNF α further increased. Importantly, the anti-inflammatory cytokine IL-10 increased only at 18 hours, suggesting the onset of the "suppression" phase. The fibrinogen and thrombin-anti-thrombin complex (TAT) had already increased at 6 hours and further increased at 18 hours, suggesting a "hypercoagulant" state (Supplementary Fig. 2d and 2e). The fibrinolytic product D-dimer only became increased at 18 hours (Supplementary Fig. 2f), consistent with its association with a more grave stage of sepsis. These results suggest that systemic inflammation had fully developed at 6 hours after sepsis onset, and reached the late grave phase associated with DIC at 18 hours. Even at this late phase, MB2mP6 treatment still had a beneficial effect.

Minor comment-

In Figure 1 f the y axis is labeled ATP but the legend says ADP.

Response: We thank the reviewer for pointing out the error and have changed the legend to ATP accordingly.

Reviewer #4 (Remarks to the Author):

This is an interesting manuscript on an important subject. It recognizes the fact that, most often, platelet and innate immune function are activated together and share molecular players. The writing, especially the introduction, is skimpy and does not explain the concept of inside-out and outside-in integrin

signaling well. Especially, in the thrombin activation. A general reader will not know that here an extremely low concentration of thrombin was used to only partially activate the integrin so that ligand binding and, thus, induction of outside-in signaling, was necessary for full platelet activation. While the experimental work presented is highly relevant to the subject at hand, the scrambled peptide should always be used as a control. From the writing, it was not clear how the peptide gets in the cells. Was it myristoylated, as the name implies MB2mP6 (Myr-FEKEKL) and in micelles, as indicated by the methods, in all experiments? How do the micelles get to the integrin tails? Their proximity to plasma membrane? This is not well explained, although it must be known. In animal experiments, does MB2mP6 inhibit Beta3 integrin also in endothelial cells?

Response: We appreciate the reviewer's constructive comments. We revised the introduction of manuscript for clarity and replaced all the saline control group's data with scrambled peptide. Indeed, the peptide inhibitor, MB2mP6 is myristoylated for membrane permeability and is formulated into a high loading peptide nanoparticle, which enhances its entry into cells (6). We thank the reviewer for the interesting question with regarding to the possible effect of MB2mP6 on endothelial cell function, which is certainly possible and we will study this in the future. Due to the scope of the current manuscript and large amount of additional experiments we needed to perform in this revision, we were not able to start a significant new project to convincingly address this question in the manuscript.

Specific comments:

1. Neutrophil functionality. While neutrophil activity can be detrimental in inflammation, it is also lifesaving in infection. From the CLP studies, the drug does not seem to have major detrimental effects, however, this should be demonstrated more convincingly both in vitro and in vivo. The effect of MB2mP6 on phagocytosis of bacteria in vitro and on the bacterial count (bacteremia) in the CLP sepsis model needs to be documented.

Response: We thank the reviewer for this insightful question. In this manuscript, we do not claim that MB2mP6 has no adverse effect. Thus far, all anti-inflammatory drugs have various degrees of the adverse effect of immune suppression. However, when inflammation becomes the major problem under septic conditions, anti-inflammatory drugs are necessary but the infection has to be controlled with antibiotics. In our experimental system, antibiotics are included in both control and MB2mP6 treatment groups, which likely minimizes the potential immunosuppressive effect of MB2mP6. However, antibiotic alone is not effective in treating sepsis. Due to large number of experiments during the past 9 months of revision, we only had the time to perform experiments testing the effect of MB2mP6 on phagocytosis of neutrophils in vitro. Our data show that MB2mP6 did not significantly affect the phagocytic function of neutrophils, which suggests that this drug may potentially be a relatively safer drug. In the future, we will perform more definitive studies to determine whether and to what degree MB2mP6 may adversely affect immune defense.

2. In vitro platelet work. If effects of MB2mP6 on platelets were not previously published, they should be shown in greater detail, i.e. representative tracing is not sufficient (several preparations and statistics should be used). Other agonists, such as ADP and high thrombin concentration, should be shown to demonstrate that inside-out signaling was not inhibited. Is alpha-granule secretion affected by the drug; P-selectin expression?

Response: We agree with the reviewer and added more data about the effects of MB2mP6 on platelet function in the revised manuscript. A range of thrombin doses was used to induce platelet aggregation and secretion. As expected, MB2mP6 partially inhibited the low-dose thrombin (less than 0.025U/mL) induced platelet aggregation and ATP and P-selectin secretion but had no effects on high-dose thrombin induced platelet aggregation (Fig. 3d, 3e and 3i). The inhibitory effects of MB2mP6 were not observed in ADP induced platelet aggregation in the presence of fibrinogen (Fig. 3h). Furthermore, we show that MB2mP6 does not affect PAR4AP-induced fibrinogen binding to platelets (Fig. 3f and 3g). These data further confirmed that MB2mP6 selectively inhibits integrin outside-in signaling but not inside-out signaling nor ligand binding function of integrin $\alpha_{IIb}\beta_3$.

3. Bleeding in animals treated with MB2mP6. Should be examined also in the stool/intestine contents, as was done for beta3-/- mice characterization. Hemorrhaging in CLP mice should be evaluated by histology in other organs other than kidney, for example, in the lung.

Response: Hemoglobin contents in the stool from septic mice treated with MB2mP6 are now shown in the new manuscript (Supplementary Fig. 6). Hemoglobin level was significantly increased in the stool samples 24 hours after CLP and was not affected by MB2mP6 treatment. These data are consistent with the results that MB2mP6 did not cause bleeding in the tail bleeding and rpA assays. Also, our data showed that MB2mP6 had no effect on hemorrhage in septic mouse lung. In contrast, our data demonstrated that MB2mP6 significantly decreased lung permeability 24 hours after CLP (Fig. 6). These data suggest that MB2mP6 in fact has therapeutic effect in reducing CLP-induced lung permeability in the lung, which is a major cause of ARDS.

4. Fibrin deposition. Does not represent thrombosis unless platelets are also in evidence. Fibrin alone may indicate vascular leakage. Platelet staining and, perhaps, VWF staining in addition to fibrinogen would make the presence of thrombosis more convincing.

Response: We agree with the reviewer's comment and added VWF staining in septic mice after MB2mP6 treatment in the revised manuscript (Supplementary Fig. 4c). We have also performed experiments to show fibrinogen levels and TAT complex in whole blood in septic mice with and without MB2mP6 treatment (Supplementary Fig. 5). The fibrinogen level was increased after CLP but not significantly altered by MB2mP6 treatment. The TAT complex was significantly increased between 6- 24 hours after CLP and suppressed in the MB2mP6 group. These data further suggest that MB2mP6 inhibited the hyper-coagulant state during sepsis.

Reference:

1. Seemann S, Zohles F, Lupp A. Comprehensive comparison of three different animal models for systemic inflammation. *J Biomed Sci.* 2017;24(1):60. doi: 10.1186/s12929-017-0370-8. PubMed PMID: 28836970; PMCID: PMC5569462.
2. Bhargava R, Altmann CJ, Andres-Hernando A, Webb RG, Okamura K, Yang Y, Falk S, Schmidt EP, Faubel S. Acute lung injury and acute kidney injury are established by four hours in experimental sepsis and are improved with pre, but not post, sepsis administration of TNF-alpha antibodies. *PLoS One.* 2013;8(11):e79037. doi: 10.1371/journal.pone.0079037. PubMed PMID: 24265742; PMCID: PMC3827109.

3. Singer G, Urakami H, Specian RD, Stokes KY, Granger DN. Platelet recruitment in the murine hepatic microvasculature during experimental sepsis: role of neutrophils. *Microcirculation*. 2006;13(2):89-97. doi: 10.1080/10739680500466343. PubMed PMID: 16459322.
4. Gill SE, Taneja R, Rohan M, Wang L, Mehta S. Pulmonary microvascular albumin leak is associated with endothelial cell death in murine sepsis-induced lung injury in vivo. *PLoS One*. 2014;9(2):e88501. doi: 10.1371/journal.pone.0088501. PubMed PMID: 24516666; PMCID: PMC3917898.
5. Speyer CL, Gao H, Rancilio NJ, Neff TA, Huffnagle GB, Sarma JV, Ward PA. Novel chemokine responsiveness and mobilization of neutrophils during sepsis. *Am J Pathol*. 2004;165(6):2187-96. doi: 10.1016/S0002-9440(10)63268-3. PubMed PMID: 15579460; PMCID: PMC1618724.
6. Pang A, Cheng N, Cui Y, Bai Y, Hong Z, Delaney MK, Zhang Y, Chang C, Wang C, Liu C, Plata PL, Zakharov A, Kabirov K, Rehman J, Skidgel RA, Malik AB, Liu Y, Lyubimov A, Gu M, Du X. High-loading Galpha13-binding EXE peptide nanoparticles prevent thrombosis and protect mice from cardiac ischemia/reperfusion injury. *Sci Transl Med*. 2020;12(552). Epub 2020/07/17. doi: 10.1126/scitranslmed.aaz7287. PubMed PMID: 32669423.

REVIEWERS' COMMENTS

Reviewer #1 (Remarks to the Author):

The authors have responded adequately to my concerns. They have also modified the text to expand on the data in the clinical trials that was confusing originally. This improves the background information for readers.

Reviewer #2 (Remarks to the Author):

The authors adequately addressed my concerns.

Reviewer #3 (Remarks to the Author):

The authors have very effectively addressed the issues raised in my initial review.
Dan Sheppard

Point-by-point responses to review comments:

Reviewer #1 (Remarks to the Author):

The authors have responded adequately to my concerns. They have also modified the text to expand on the data in the clinical trials that was confusing originally. This improves the background information for readers.

Response: We thank the reviewer for the positive comments.

Reviewer #2 (Remarks to the Author):

The authors adequately addressed my concerns.

Response: We thank the reviewer for the positive comments.

Reviewer #3 (Remarks to the Author):

The authors have very effectively addressed the issues raised in my initial review.

Response: We thank the reviewer for encouragement.